# Interaction between Sovereign Quanto Credit Default Swap Spreads and Currency Options

Masaru Tsuruta [†] [ID]

SBI Shinsei Bank, Limited, Tokyo 103-8303, Japan; tsurutamasaru@gmail.com
† Current address: 303 GraceMaison-Jimbocho, 1–50, Kanda-Jimbo-cho, Chiyoda-ku, Tokyo 101-0051, Japan.

**Abstract:** This study analyzes the term structures of sovereign quanto credit default swap (CDS) spreads and currency options, which are driven by anticipated currency depreciation risk following sovereign credit default (Twin Ds). We develop consistent pricing models for these instruments using a jump-diffusion stochastic volatility model, which allows us to decompose the term structure into the risk components. We find a common risk factor between the intensity process of sovereign credit risk and the stochastic volatility of the exchange rate, and the depreciation risk mainly captures the dependence structure between these markets during periods of high market stress in the Eurozone countries. Depreciation risk is an important component of sovereign quanto CDS spreads and is evident in the negative slope of the volatility smile in the currency option market.

**Keywords:** sovereign credit default swaps; currency option; sovereign credit risk; jump-diffusion stochastic volatility model; depreciation risk

## 1. Introduction

During the European sovereign debt crisis, which began in 2009[1], the sovereign credit default swap (CDS) spreads of Eurozone countries fluctuated widely, and the exchange rate of the euro (EUR) against the U.S. dollar (USD) depreciated and remained volatile. During the same period, the currency option of EURUSD—which reflects the anticipated change in the exchange rate—increased. Additionally, the negative slope of the volatility smile for EURUSD became steeper, implying that the skewness of the anticipated currency return distribution increased. Hence, the sovereign CDS spread, currency option implied volatility, and slope of the volatility smile covaried. Thus, sovereign credit concerns tend to be associated with the co-movement of the sovereign credit risk and currency option markets.

Sovereign defaults are typically accompanied by large exchange rate devaluation, a phenomenon commonly referred to as the Twin Ds (default and devaluation; see Na et al. 2018). Na et al. (2018) report that a large depreciation coinciding with a sovereign default is not followed by an increase in the depreciation rate. The sovereign quanto CDS spread, which is the difference between different currency-denominated CDS spreads of the same underlying entity, provides information on the dependence structure between the exchange rate and sovereign credit risk, including an anticipated large depreciation coinciding with a sovereign default (see Ehlers and Schönbucher 2004; Lando and Bang Nielsen 2018). In addition to the risks associated with currency depreciation, Carr and Wu (2007) and Della Corte et al. (2022) mention that uncertainty in the currency market caused by sovereign credit-related concerns could increase currency option implied volatility.

Several studies examine the relationship between the term structures of sovereign CDSs and currency markets. Carr and Wu (2007) investigate the co-movement of the USD-denominated sovereign CDS spread and the currency option implied volatility in Mexico and Brazil without the quanto spread. Therefore, they do not consider a market-implied large depreciation coinciding with a sovereign default. Augustin et al. (2020) estimate the EUR depreciation risk at credit events using a joint valuation model for sovereign

quanto CDS spreads and spot/forward exchange rates, but their model does not include the dependence structure and the correlation risk in the quanto spreads. Lando and Bang Nielsen (2018) study the impact of the depreciation risk and the correlation risk on sovereign quanto CDS spreads. However, the influence of depreciation risk on the currency market and the dependence structure between the sovereign credit risk intensity process and the stochastic volatility process of the exchange rate are not considered in their model. To complement these studies, this study aims to investigate the interactions of term structures with different currency-denominated sovereign CDS spreads and currency options by considering the depreciation risk and dependence structures, including the correlation risk, using a consistent model in both markets.

We use USD- and local currency-denominated sovereign CDS spreads and the currency option implied volatilities of Eurozone countries and the United Kingdom (UK) for two economic phases: the European sovereign debt crisis and the post-crisis period of economic expansion. We examine each country at one-, three-, and five-year maturities. For currency options, we use the delta-neutral straddle implied volatilities, 25-delta risk reversals ($RR$), and 25-delta butterfly spreads ($BF$) at one-month, one-year, and five-year maturities. Following the market convention, currency options are quoted in terms of a delta-neutral straddle implied volatility, 25-delta risk reversals, and 25-delta butterfly spreads at each maturity. The delta-neutral straddle implied volatility is the implied volatility of puts and calls with the same delta value. We observe the delta-neutral straddle implied volatility as the at-the-money implied volatility ($ATMV$). The 25-delta risk reversal is the difference in implied volatilities based on the Black–Scholes formula between a 25-delta call option ($IV(25dCall)$) and a 25-delta put option ($IV(25dPut)$), which is:

$$RR = IV(25dCall) - IV(25dPut). \tag{1}$$

Therefore, risk reversal represents the slope of the implied volatility against moneyness. The 25-delta butterfly spread is the difference between the average implied volatility based on the Black–Scholes formula at the two 25-delta and delta-neutral straddle implied volatilities:

$$BF = (IV(25dCall) + IV(25dPut))/2 - ATMV. \tag{2}$$

Thus, the butterfly spread represents the average curvature of the implied volatility against moneyness.

We develop consistent pricing models for these instruments using a jump-diffusion stochastic volatility model, which allows us to decompose the term structures into their risk components. The risk components include the exchange rate's depreciation risk in response to a sovereign credit event, the correlation risk, and the common risk factor between the sovereign credit and currency option markets. For tractability, we extend the model of Carr and Wu (2007). We capture depreciation risk following a sovereign credit event by using the jump process in the exchange rate process. To represent the link between currency option implied volatility and sovereign credit-related concerns, we assume that the intensity process of the sovereign credit event comprises the stochastic volatility process. Additionally, we assume that the stochastic volatility is correlated with the exchange rate.[2] By summarizing these structures, we can also capture the correlation risk between the intensity process and the exchange rate. We then apply these processes to currency options and different currency-denominated sovereign CDS spreads. Accordingly, this model allows us to decompose the term structures into their risk components. Although the correlation values between CDS spreads and currency option implied volatility are positive when based on historical dynamics, we find that the estimated intensity process does not comprise the stochastic volatility process. The estimated mean reversion parameters of these processes differ, and thus their effects on the slope of the term structure differ. In the one-factor stochastic volatility model such as Carr and Wu (2007), stochastic volatility cannot capture the common factor for sovereign CDS markets, and each process mainly corresponds to each market. Therefore, we introduce an idiosyncratic risk process in

stochastic volatility to capture the unique risk in the currency market, while keeping the first stochastic volatility as a common factor with sovereign credit risk. We also estimate a two-factor stochastic volatility model in this manner.

Contrary to the findings presented in Carr and Wu (2007), our analysis using the quanto CDS spreads reveals that a two-factor stochastic volatility model provides a more accurate representation of the relationship between sovereign CDS spreads and currency options, particularly in the context of the European sovereign debt crisis. Our study shows how, during the European sovereign debt crisis, the interaction between sovereign CDS spreads and currency options in Eurozone countries is significantly influenced by two principal risk components. The common risk factor is identified as a primary determinant of CDS spreads, with its impact being especially pronounced in Italy and Spain during times of high market stress. During the same period, depreciation risk following a sovereign credit event drives quanto CDS spreads and the negative slope of the volatility smile in currency options. The common risk factor also influences currency option implied volatility and accentuates the steepness of the volatility smile's negative slope. These findings contribute to a deeper understanding of market dynamics under stress. Interestingly, our analysis indicates that the effect of correlation risk on quanto CDS spreads is comparatively minor, diverging from the findings of Lando and Bang Nielsen (2018).

Overall, our study enriches the literature by providing nuanced insights into the dynamics between sovereign quanto CDS spreads and currency options. It highlights the crucial role of common risk factors and depreciation risk during periods of financial distress, particularly within the Eurozone amidst the European sovereign debt crisis. Through this contribution, we offer a comprehensive framework for understanding and modeling the interactions between sovereign credit and currency risks, thereby paving the way for further research in this domain.

The rest of the paper is organized as follows. In Section 2, we briefly review the related literature. Section 3 describes the pricing model of the CDS spread and the currency option based on the one-factor stochastic volatility model. Section 4 discusses the data and the estimation approach. Additionally, we present the results of the one-factor stochastic volatility model. Section 5 provides the two-factor stochastic volatility model and its results. Section 6 concludes the study.

## 2. Related Literature

Carr and Wu (2007) study the co-movement between the term structures of sovereign CDSs and currency option implied volatility, using stochastic volatility with a jump process for the exchange rate process in Mexico and Brazil. They show that the historical average default intensity is lower than those estimated from sovereign CDS spreads under the risk-neutral measure. However, this analysis estimates the jump size solely from currency option markets. Therefore, the jump size is not directly linked to sovereign credit events and does not primarily focus on the depreciation risk. By using the information implied in the sovereign quanto CDS spreads, which are not available before August 2010, we can then estimate the jump size following a sovereign credit event and the importance of depreciation risk in both markets, while maintaining consistency between the markets. Furthermore, they pay little attention to the impact of risk factors on each instrument. By contrast, our study presents the impact of each risk factor.

Augustin et al. (2020) investigate the risk premium of currency depreciation following a sovereign credit event. They estimate the EUR depreciation risk at a credit event using a joint valuation model for the term structures of sovereign quanto CDS spreads and spot/forward exchange rates. They demonstrate that the risk premium of currency depreciation following a credit event is large, assuming that an extreme change in quanto CDS spreads is a proxy for credit events under a physical probability based on a rough assumption. They consider the relationship between sovereign risk and currency spot/forward only through depreciation risk. Consequently, they do not consider the correlation risk and co-movement risk between the intensity process and currency market, except for the

depreciation risk following sovereign credit events. Our model considers the risks between the exchange rate and intensity process using a currency option. We also investigate the impact of currency depreciation on the skewness of the currency returns.

Lando and Bang Nielsen (2018) investigate the contribution of depreciation risk and covariance risk between the term structures of the quanto CDS spread and the impact of the quanto effect on the yield spread between Eurozone sovereign bonds issued in EUR and USD. They show that covariance risk in the quanto CDS spread is important in times of distress and that the quanto yield spreads are related to the quanto effect estimated from the quanto CDS spread. However, they do not focus on the interactions between the two markets. Their model does not include the impact of sovereign credit risk, especially the depreciation risk, on the currency option. In contrast to Lando and Bang Nielsen (2018), we focus on the interaction between the currency option and the sovereign quanto CDS markets and apply a consistent model to both markets.

A growing body of literature investigates the interaction between exchange rates and sovereign credit risk. Della Corte et al. (2022) show that sovereign credit risk is correlated with currency excess return and its volatility and skewness. Della Corte et al. (2022) exploit sovereign quanto CDS spreads to predict the EUR/USD exchange rate returns. Du and Schreger (2016) study credit risks for local currencies in emerging countries using a cross-currency swap and local currency-denominated sovereign bond yields. Mano (2013) and De Santis (2015) estimate the depreciation risk using quanto CDS spread data. Although they note that foreign currency-denominated sovereign CDSs have exchange rate and depreciation risks following sovereign credit events, they do not use currency options data. We argue that the estimated depreciation risk is lower when the model includes correlation risk relative to when it does not. Ehlers and Schönbucher (2004) and Brigo et al. (2019) propose the approach of linking the foreign exchange rate and intensity process by considering an affine jump-diffusion model for an exchange rate process in which the jump occurs at the credit event. Monfort et al. (2021) investigate the frailty and contagion phenomena in sovereign CDS spreads and develop a model of sovereign quanto CDS spreads in the Eurozone using an autoregressive gamma process.

## 3. The Pricing Model

### 3.1. Diffusion Process

We construct a model to capture the dependence between different currency-denominated sovereign CDS spreads and currency options. We assume that the pre-credit event exchange rate process follows a stochastic volatility model.[3] This model allows us to capture the shape of the volatility smile. Additionally, we assume that a jump in the exchange rate process coincides with a sovereign credit event.

We denote the exchange rate between the foreign and domestic currencies as $X_t \geq 0$. $X_t$ is the value of one foreign currency unit expressed in the domestic currency unit at a time $t$. In our model, we refer to the USD as the domestic currency and the currency of the country referenced by the sovereign CDS as the foreign currency. This is because the benchmark sovereign CDS is denominated in the USD, and we use the dollar price of its currency for the currency option. We denote the short-term interest rate processes for the domestic and foreign currencies as $r_t^d$ and $r_t^f$, respectively. The complete filtered probability space, $(\Omega, \mathcal{G}, (\mathcal{G}_t), \mathbb{Q})$, is fixed. We denote this information as $\mathcal{G}_t = \mathcal{F}_t \vee \mathcal{H}_t$ at a time $t$, where $\mathcal{F}_t$ is generated by the standard Brownian motion $w$ and $\mathcal{H}_t$ is generated by the compound Poisson process $J_t$ associated with a sovereign credit event. $J_t$ represents the foreign currency's depreciation against the domestic currency following a sovereign credit event. We express a domestic measure as $\mathbb{Q}_d$ instead of just $\mathbb{Q}$. $J_t$ is associated with the jump size $\delta_x^d$, which represents the depreciation ratio. We assume that $\delta_x^d := 1 - e^{-q}$ and $q$ have a normal distribution with a mean $\mu_j$ and a variance $v_j$, which are constant parameters under $\mathbb{Q}_d$. The expectation of $\delta_x^d$ is denoted as $\hat{\delta}_x^d := 1 - e^{-\mu_j + v_j/2}$. The intensity process of the jump process is denoted as $\lambda_t^d$ and is a stochastic process.

First, we consider a single currency adopted by a single country. Under the domestic risk-neutral measure, $\mathbb{Q}_d$, $X_t$ satisfies a stochastic differential equation (SDE) of the following form:

$$\frac{dX_t}{X_t} = (r_t^d - r_t^f)dt + \sqrt{v_t}dw_{x,t}^d - (\delta_x^d dJ_t - \hat{\delta}_x^d \lambda_t^d dt), \tag{3}$$

$$dv_t = k_v^Q(\theta_v^Q - v_t)dt + \sigma_v \sqrt{v_t}dw_{v,t}^d, \tag{4}$$

$$E[dw_{x,t}^d dw_{v,t}^d] = \rho dt,$$

where $v_t$ is the instantaneous conditional variance obtained using a stochastic process and follows a mean-reverting square-root process.

To capture the dynamics between sovereign CDSs and currency options, the intensity process is assumed to consist of the stochastic volatility process $v_t$ and the idiosyncratic risk process $z_t$. We capture the relationship between the intensity process and stochastic volatility process through the coefficient $\beta(\geq 0)$. Under $\mathbb{Q}_d$, we assume that the dynamics of the intensity process adhere to the following equations:

$$\lambda_t^d = \beta v_t + z_t, \tag{5}$$

$$dz_t = k_z^Q(\theta_z^Q - z_t)dt + \sigma_z \sqrt{z_t}dw_t^z, \tag{6}$$

$$E[dw_{x,t}^d dw_t^z] = 0,$$

where $z_t$ is the mean-reverting square-root process. Following Proposition 2 of Ehlers and Schönbucher (2004) and the Girsanov theorem, we derive the diffusion process under a foreign risk-neutral measure, $\mathbb{Q}_f$, as follows:

$$\lambda_t^f = (1 - \hat{\delta}_x^d)\lambda_t^d, \tag{7}$$

$$dv_t = (k_v^Q\theta_v^Q - (k_v^Q - \rho\sigma_v)v_t)dt + \sigma_v \sqrt{v_t}dw_{v,t}^f, \tag{8}$$

$$dz_t = k_z^Q(\theta_z^Q - z_t)dt + \sigma_z \sqrt{z_t}dw_t^z. \tag{9}$$

Next, we consider the case in which multiple countries adopt the same currency as a currency union. This setting is intended to apply to Eurozone countries. There are $N$ $(i = 1, \cdots, N)$ countries, and the intensity process of each country is denoted as $\lambda_{i,t}^d$. We extend Equations (3) and (4) to the exchange rate process. Under the domestic risk-neutral measure $\mathbb{Q}_d$, the exchange rate process $X_t$ of the currency adopted by multiple countries is assumed to satisfy the SDE as follows:

$$\frac{dX_t}{X_t} = (r_t^d - r_t^f)dt + \sqrt{v_t}dw_{x,t}^d - \sum_{i=1}^{N}(\delta_{x,i}^d dJ_{i,t} - \hat{\delta}_{x,i}^d \lambda_{i,t}^d dt), \tag{10}$$

$$dv_t = k_v^Q(\theta_v^Q - v_t)dt + \sigma_v \sqrt{v_t}dw_{v,t}^d, \tag{11}$$

$$E[dw_{x,t}^d dw_{v,t}^d] = \rho_i dt.$$

We also assume that the intensity process for each country consists of the stochastic volatility process $v_t$ and the idiosyncratic risk process $z_{i,t}$. We capture the relationship between the intensity process and stochastic volatility process through the coefficient $\beta_i(\geq 0)$ for each country. Extending Equations (5) and (6), we assume that the dynamics of the intensity process under $\mathbb{Q}_d$ are assumed to adhere to the following equations:

$$\lambda_{i,t}^d = \beta_i v_t + z_{i,t}, \tag{12}$$

$$dz_{i,t} = k_{z_i}^Q(\theta_{z_i}^Q - z_{i,t})dt + \sigma_{z_i} \sqrt{z_{i,t}}dw_{i,t}^z, \tag{13}$$

$$E[dw_{x,t}^d dw_{i,t}^z] = 0.$$

Similarly, by extending Equations (7) to (9), we can derive the diffusion process under foreign risk-neutral measure $\mathbb{Q}_f$ as follows:

$$
\begin{align}
\lambda_{i,t}^f &= (1 - \hat{\delta}_{x,i}^d)\lambda_{i,t}^d, \tag{14} \\
dv_t &= (k_v^Q \theta_v^Q - (k_v^Q - \rho\sigma_v)v_t)dt + \sigma_v\sqrt{v_t}dw_{v,t}^f, \tag{15} \\
dz_{i,t} &= k_{z_i}^Q(\theta_{z_i}^Q - z_{i,t})dt + \sigma_{z_i}\sqrt{z_{i,t}}dw_t^{z_i}. \tag{16}
\end{align}
$$

In our model, we use the overnight indexed swap (OIS) as an alternative variable to the interest rate and assume that the risk-free rate is not stochastic.

### 3.2. Domestic Currency-Denominated Sovereign CDS Pricing Model

This subsection describes the pricing formula for domestic currency-denominated CDS spreads. We first present a simple example before discussing the main subject. Let us consider that USD 1 million of Italian CDS protection is held. When buying CDS protection, the exchange rate is assumed to be 1 USD/EUR. Then, we consider the time at which a credit event occurs in Italy and assume that the recovery rate is 50%. Additionally, we assume that the exchange rate depreciates, and its price is 0.5 USD/EUR (=2 EUR/USD). In this case, the protection holder requires USD 0.5 million. The holder of a USD-denominated Italian CDS can obtain EUR 1 million (=2 EUR/USD × 0.5 million USD). The holder of the EUR-denominated Italian CDS, the notional value of which is EUR 1 million, can obtain EUR 0.5 million. There is a difference in the payment values. The pricing formula of the USD-denominated CDS[4] for Italy is considered in this subsection.

The value of the CDS spread at a time $t$, the maturity of which occurs at $t + m$, is denoted as $CDS_t(m)$. Following Longstaff et al. (2005) and Carr and Wu (2007), the sovereign CDS spread denominated in the domestic currency of a foreign country is given by

$$
S^d(m, \lambda_t^d) = \frac{(1 - R)\int_t^{t+m} E^{\mathbb{Q}_d}\left[e^{-\int_t^s r_u^d + \lambda_u^d du}\lambda_s^d \middle| \mathcal{F}_t\right]ds}{\Delta t \sum_{k=1}^{m/\Delta t} E^{\mathbb{Q}_d}\left[e^{-\int_t^{t+k/\Delta t} r_u^d + \lambda_u^d du} \middle| \mathcal{F}_t\right]}. \tag{17}
$$

$\lambda_t^d$ follows Equation (5) with Equations (4) and (6) for the case where a single currency is adopted by a single country or Equation (12) with Equations (11) and (13) for the case where a single currency is adopted by multiple countries. Here, we assume that the recovery model is assumed to be the recovery of face value. Under the recovery of face value, if a credit event occurs at $\tau(\leq T)$, the bond holder receives a recovery payment of the size $(1 - R)$ immediately at the time of the credit event $\tau$. Recovery of face value is the recovery model closest to becoming a legal practice.[5] The recovery rate is set to 40%,[6] which is a typical assumption for the market convention rate for standard Western European sovereign in practice. We can analytically evaluate the domestic currency-denominated sovereign CDS spread formula, as shown in Appendix A.

### 3.3. Foreign Currency-Denominated Sovereign CDS Pricing Model

Next, we describe the pricing formula for foreign currency-denominated sovereign CDS spread. The currency of the country is referenced by the sovereign CDS as the foreign currency to maintain a consistent notation. Thus, the pricing formula considered in this subsection is applied to EUR-denominated CDSs, for example, in Italy.

Foreign currency-denominated sovereign CDS spreads are given by the following equation:

$$
S^f(m, \lambda_t^f) = \frac{(1-R) \int_t^{t+m} E^{\mathbb{Q}_f}\left[ e^{-\int_t^s r_u^f + \lambda_u^f du} \lambda_s^f \middle| \mathcal{F}_t \right] ds}{\Delta t \sum_{k=1}^{m/\Delta t} E^{\mathbb{Q}_f}\left[ e^{-\int_t^{t+k/\Delta t} r_u^f + \lambda_u^f du} \middle| \mathcal{F}_t \right]}. \tag{18}
$$

$\lambda_t^f$ follows Equation (7) with Equations (8) and (9) for the case in which a single currency is adopted by a single country or Equation (14) with Equations (15) and (16) for the case where a single currency is adopted by multiple countries. We analytically evaluate the foreign currency sovereign CDS spread formula, as shown in Appendix A.

### 3.4. Currency Option

The value of a European call option on the dollar price of a foreign currency (e.g., the GBPUSD and EURUSD, where GBP is the pound sterling) is:

$$
c(m) = E^{\mathbb{Q}_d}[e^{-\int_t^{t+m} r_u^d du}(X_T - K)^+ | \mathcal{F}_t], \tag{19}
$$

where $K$ is the strike price and $T = t + m$ is the expiry date. We can solve for this value using the generalized Fourier transform of the log of currency:

$$
\phi(u) = E^{\mathbb{Q}_d}[e^{iu \ln X_t} | \mathcal{F}_t].
$$

We provide a detailed solution in Appendix B. To calculate the currency option price from the generalized Fourier transform of the log currency, we use the COS method (Fang and Oosterlee 2009), which is based on the Fourier-cosine series for solving inverse Fourier integrals. Thereby, we can calculate the value of the currency option with the diffusions in Equations (3)–(6) or (10)–(13).

## 4. Model Estimation

In this section, we estimate the model using domestic and foreign currency-denominated CDS spreads and currency option prices.

### 4.1. Data

The data include weekly domestic and foreign currency-denominated CDS[7] data at one-, three-, and five-year maturities.[8] We also include weekly currency option prices for one-month, one-year, and five-year maturities. Although Carr and Wu (2007) use a one-year or shorter maturity, we include a five-year maturity that overlaps with the maturities of sovereign CDS spreads to capture the joint dynamics between both instruments using the common risk factor. Our data include the UK and four Eurozone countries: Spain, Italy, Ireland, and Portugal.[9] For the UK, we apply the model to the currency option for the dollar price of the GBP, as well as to USD- and GBP-denominated CDSs for a single country. For Eurozone countries, we simultaneously apply the model to a currency option for the USD price of the EUR, as well as to USD- and EUR-denominated CDSs for the four Eurozone countries.

The data cover 436 weeks from 25 August 2010 to 26 December 2018, at a weekly frequency (we use data for the Wednesday of each week). The currency option implied volatility and OIS data are obtained from Bloomberg, and the CDS data are from IHS Markit. Although different currency-denominated CDS spreads are required in our model, CDS spread quotes for different currencies are not available before August 2010. Therefore, we collect our sample data beginning in August 2010. Additionally, to investigate the two financial phases, we divide the observed period into the European sovereign crisis period (Period 1: from 25 August 2010 to 31 December 2014) and the post-crisis period (Period

2: from 7 January 2015, to 26 December 2018). The UK's withdrawal from the European Union (Brexit) after the European Union Referendum Act of 2015 has become a concern in the market. This event occurred during Period 2.

For the currency option implied volatility, we use the formulas in Equations (1) and (2) to calculate the implied volatilities at the two deltas based on the Black–Scholes formula. Additionally, we convert these implied volatilities into option prices.

Table 1 shows the cross-correlation between sovereign CDS spreads at one-year and five-year maturities, as well as currency options at one-month, one-year, and five-year maturities in weekly differences. For Period 1, the correlation estimates between the CDS spreads at both maturities and the at-the-money implied volatilities at all maturities are positive and range from 0.19 to 0.55. The estimates for Italy and Spain are higher than those for the other countries. Further, the correlation estimates between the CDS spreads at both maturities and the risk reversals at one-month and one-year maturities are negative and range from $-0.43$ to $-0.22$. In contrast, the correlation estimates between the CDS spreads at both maturities and the butterfly spreads at all maturities are close to zero or are small positive values.

**Table 1.** Correlations between CDS spreads, exchange rates, and currency option implied volatilities. The table shows the sample correlation between weekly changes in USD-denominated CDS spreads for the UK, Spain (ES), Italy (IT), Ireland (IE), and Portugal (PT); the currency option implied volatility of GBPUSD for the UK; and the EURUSD for the Eurozone countries at each maturity and slope of the term structures. The sample period for Period 1 is from 25 August 2010 to 31 December 2014, with a weekly frequency. The sample period for Period 2 is from 7 January 2015 to 26 December 2018, with a weekly frequency.

| | | ATM | | | RR | | | BF | | |
|---|---|---|---|---|---|---|---|---|---|---|
| | | 1M | 1Y | 5Y | 1M | 1Y | 5Y | 1M | 1Y | 5Y |
| CDS Spread, Period 1 | | | | | | | | | | |
| UK | 1Y | 0.28 | 0.28 | 0.19 | −0.24 | −0.30 | −0.04 | 0.26 | 0.22 | 0.00 |
| | 5Y | 0.31 | 0.32 | 0.21 | −0.29 | −0.34 | −0.09 | 0.29 | 0.17 | 0.01 |
| ES | 1Y | 0.44 | 0.49 | 0.47 | −0.38 | −0.33 | −0.03 | 0.12 | 0.05 | 0.06 |
| | 5Y | 0.45 | 0.48 | 0.46 | −0.39 | −0.33 | −0.04 | 0.11 | 0.04 | 0.08 |
| IT | 1Y | 0.53 | 0.55 | 0.52 | −0.40 | −0.34 | −0.04 | 0.14 | 0.10 | 0.10 |
| | 5Y | 0.55 | 0.54 | 0.53 | −0.43 | −0.37 | −0.05 | 0.17 | 0.11 | 0.12 |
| IE | 1Y | 0.32 | 0.29 | 0.26 | −0.28 | −0.31 | −0.09 | 0.14 | 0.14 | 0.03 |
| | 5Y | 0.35 | 0.29 | 0.26 | −0.31 | −0.33 | −0.06 | 0.18 | 0.16 | 0.03 |
| PT | 1Y | 0.32 | 0.28 | 0.23 | −0.22 | −0.25 | 0.00 | 0.16 | 0.18 | −0.08 |
| | 5Y | 0.36 | 0.32 | 0.27 | −0.24 | −0.24 | 0.00 | 0.18 | 0.19 | −0.01 |
| CDS Spread, Period 2 | | | | | | | | | | |
| UK | 1Y | 0.17 | 0.33 | 0.31 | −0.08 | −0.22 | −0.11 | 0.18 | 0.21 | −0.10 |
| | 5Y | 0.10 | 0.38 | 0.39 | −0.09 | −0.26 | −0.17 | 0.14 | 0.29 | −0.10 |
| ES | 1Y | 0.22 | 0.19 | 0.18 | −0.11 | −0.21 | −0.21 | 0.18 | 0.28 | 0.03 |
| | 5Y | 0.20 | 0.19 | 0.19 | −0.12 | −0.24 | −0.19 | 0.21 | 0.32 | −0.02 |
| IT | 1Y | 0.20 | 0.25 | 0.25 | −0.15 | −0.34 | −0.17 | 0.19 | 0.28 | 0.03 |
| | 5Y | 0.23 | 0.25 | 0.26 | −0.14 | −0.31 | −0.12 | 0.22 | 0.31 | 0.00 |
| IE | 1Y | 0.17 | 0.15 | 0.13 | −0.10 | −0.23 | −0.04 | 0.07 | 0.24 | 0.09 |
| | 5Y | 0.25 | 0.26 | 0.23 | −0.15 | −0.30 | −0.08 | 0.18 | 0.32 | 0.02 |
| PT | 1Y | 0.24 | 0.19 | 0.17 | −0.08 | −0.23 | −0.05 | 0.20 | 0.32 | 0.01 |
| | 5Y | 0.23 | 0.19 | 0.18 | −0.08 | −0.21 | −0.05 | 0.21 | 0.31 | 0.01 |

Overall, the absolute values of the correlation estimates for the Eurozone countries in Period 2 are almost the same or smaller than those in Period 1. During Period 2, the UK engaged in Brexit discussions. The correlation estimates between the CDS spreads at both maturities and the at-the-money implied volatilities at one-year and five-year maturities for Period 2 are the same or higher compared to those for Period 1. Thus, local political/financial concerns cause the correlation between sovereign CDS spreads and at-the-money implied volatilities and risk reversal in the currency market to be higher than that during the calm period, in which no crises or Brexit discussions.

*4.2. The State Space Model*

To estimate the parameters and processes of the model, we recast it within the framework of a state space model. First, we describe the measurement equation, which comprises the pricing formulas in Equations (17)–(19). The measurement equations are

$$
\begin{aligned}
y_{CDS,d,t,m_c} &= S^d(m_c, v_t, z_t) + \epsilon_{1,m_c,t}, \; m_c = 1,3,5 \text{ years}, & (20) \\
y_{CDS,f,t,m_c} &= S^f(m_c, v_t, z_t) + \epsilon_{2,m_c,t}, \; m_c = 1,3,5 \text{ years}, & (21) \\
\frac{y_{opt,t,m_o,K}}{vega_{t,m_o,K}} &= \frac{c(m_o, K, v_t, z_t)}{vega_{t,m_o,K}} + \epsilon_{3,m_o,t}, \; m_o = 1 \text{ month}, 1, 5 \text{ years}, \\
& \qquad\qquad K \text{ at three deltas}, & (22)
\end{aligned}
$$

$$
\epsilon_{1,m_c,t} \sim i.i.d.N(0,\tau_1), \; \epsilon_{2,m_c,t} \sim i.i.d.N(0,\tau_1), \; \epsilon_{3,m_o,t} \sim i.i.d.N(0,\tau_2),
$$

where $y_{CDS,d,t,m_c}$ denotes the observed variables of domestic currency-denominated sovereign CDS spreads with a maturity $m_c = 1,3,5$ years at a time $t$; $y_{CDS,f,m_c,t}$ denotes the observed variable of foreign currency-denominated sovereign CDS spreads with a maturity $m_c = 1,3,5$ years at a time $t$; and $y_{opt,t}$ denotes the observed variable of currency option prices with a maturity $m_o = 1$ month, 1 year, 5 years and including a 25-delta call, a 25-delta put, and the delta-neutral straggle at a time $t$. Pricing errors $\epsilon_{l,m,t}$ are independent and identically distributed according to the Gaussian distribution. For Eurozone countries, $y_{CDS,d,t,m_c}$ and $y_{CDS,f,t,m_c}$ consist of the sovereign CDS spreads of the four countries. To avoid overfitting to one instrument for the currency options, we represent the currency option prices scaled by their Black–Scholes vega, $vega_{t,m_o,K}$.[10] These measurement equations are not linear.

Next, we describe the transition equation. To estimate the time series of the idiosyncratic risk process and the stochastic volatility process, we assume that these processes follow the square-root process under the physical measure:

$$
\begin{aligned}
dz_{i,t} &= k_{z_i}^P(\theta_{z_i}^P - z_{i,t})dt + \sigma_{z_i}\sqrt{z_{i,t}}dw_t^{z_i,P}, \\
dv_t &= k_v^P(\theta_v^P - v_t)dt + \sigma_v\sqrt{v_t}dw_{v,t}^P.
\end{aligned}
$$

To derive the dynamics under a physical measure, the market price of risk, $\eta_t$, for the idiosyncratic risk process, $z_{i,t}$, is assumed to be: $\eta_t = \frac{\psi_{i,0}}{\sqrt{z_{i,t}}} + \psi_{i,1}\sqrt{z_{i,t}}$, where $k_{z_i}^Q\theta_{z_i}^Q = k_{z_i}^P\theta_{z_i}^P - \psi_{i,0}\sigma_{z_i}, k_{z_i}^Q = k_{z_i}^P + \sigma_{z_i}\psi_{i,1}$.[11] For the stochastic volatility process $v_t$, the market price of risk is assumed to follow a formula similar to the idiosyncratic risk process $z_{i,t}$. The conditional distributions of the idiosyncratic risk and stochastic volatility processes are non-central Chi-square distributions and are not Gaussian. We treat these as if the state variables were conditionally normally distributed. Duan and Simonato (1999) and Chen and Scott (2003) suggest an approximation methodology[12] for square-root-type state variables.

Because Equations (20)–(22) are nonlinear, we apply the unscented Kalman filter for the estimation. We jointly estimate the model parameters using the term structures of USD- and local currency-denominated CDS spreads and the term structures of currency options. We jointly estimate the model parameters for the four Eurozone countries. We construct a quasi-log-likelihood function, $L(\Theta)$, for this state space model. The quasi-maximum likelihood estimator is $\hat{\Theta} = arg\max_\Theta L(\Theta)$. In addition, we restrict $k_{z_i}^P, \theta_{z_i}^P, k_v^P$, and $\theta_v^P$ to be

positive to allow for the square-root process. The risk-neutral measure parameters of the square-root process are unconstrained. The positivity of the filtered square-root process is ensured by setting the joint likelihood of the entire time series to zero whenever the filtered expectation of the square-root process is negative, following Chen and Scott (2003).

*4.3. Empirical Results*

4.3.1. Estimated Parameters

Panel A of Table 2 presents the estimated parameters and their asymptotic standard errors for the UK. The signs of the risk-neutral mean reversion parameters $k_v^Q$ and $k_z^Q$ differ, where $k_v^Q$ is positive and $k_z^Q$ is negative.[13] Thus, the risk-neutral behavior of the stochastic volatility $v_t$ and idiosyncratic risk $z_t$ are different. This pattern is indicated and discussed by Carr and Wu (2007).

**Table 2.** Estimated parameters from the one-factor stochastic volatility model for the UK and Eurozone countries. The Eurozone countries are Spain (ES), Italy (IT), Ireland (IE), and Portugal (PT). The table shows the quasi-maximum likelihood estimator. The asymptotic standard errors are in parentheses. The asymptotic standard errors are based on the inverse of the information matrix computed from the Hessian matrix and the gradient vector for the log-likelihood function. In Panel A, Period 1 is from 25 August 2010, to 31 December 2014, with a weekly frequency. In Panel B, Period 2 is from 7 January 2015, to 26 December 2018, with a weekly frequency.

| | Panel A : UK | | Panel B : Eurozone Countries | | | | | | | | | |
|---|---|---|---|---|---|---|---|---|---|---|---|---|
| | Period 1 | Period 2 | Period 1 | | | | | Period 2 | | | | |
| | | | | Country | | | | | Country | | | |
| | | | | ES | IT | IE | PT | | ES | IT | IE | PT |
| $k_v^Q$ | 0.109 | 3.281 | 0.667 | | | | | 1.541 | | | | |
| | (0.003) | (0.009) | (0.004) | | | | | (0.006) | | | | |
| $k_v^Q\theta_v^Q$ | 0.003 | 0.031 | 0.005 | | | | | 0.010 | | | | |
| | (0.002) | (0.010) | (0.003) | | | | | (0.002) | | | | |
| $\sigma_v$ | 0.060 | 0.201 | 0.069 | | | | | 0.118 | | | | |
| | (0.000) | (0.001) | (0.000) | | | | | (0.001) | | | | |
| $k_v^P$ | 0.846 | 4.189 | 0.566 | | | | | 0.064 | | | | |
| | (0.003) | (0.152) | (0.046) | | | | | (0.016) | | | | |
| $\theta_v^P$ | 0.008 | 0.010 | 0.010 | | | | | 0.000 | | | | |
| | (0.000) | (0.000) | (0.000) | | | | | (0.000) | | | | |
| $\rho$ | −0.499 | −0.562 | −0.404 | | | | | −0.060 | | | | |
| | (0.001) | (0.004) | (0.000) | | | | | (0.001) | | | | |
| $\tau_{cur}$ | 0.723 | 0.857 | 0.582 | | | | | 0.573 | | | | |
| | (0.008) | (0.017) | (0.011) | | | | | (0.011) | | | | |
| $k_z^Q$ | −0.400 | −0.345 | | −0.108 | −0.051 | 0.000 | −0.031 | | −0.286 | −0.249 | −0.355 | −0.596 |
| | (0.004) | (0.026) | | (0.001) | (0.001) | (0.000) | (0.006) | | (0.003) | (0.014) | (0.011) | (0.002) |
| $k_z^Q\theta_z^Q$ | 0.000 | 0.000 | | 0.005 | 0.008 | 0.007 | 0.018 | | 0.001 | 0.004 | 0.001 | 0.002 |
| | (0.001) | (0.004) | | (0.003) | (0.006) | (0.002) | (0.009) | | (0.002) | (0.003) | (0.005) | (0.001) |
| $\sigma_z$ | 0.036 | 0.026 | | 0.152 | 0.189 | 0.270 | 0.403 | | 0.076 | 0.254 | 0.030 | 0.265 |
| | (0.000) | (0.003) | | (0.000) | (0.001) | (0.001) | (0.004) | | (0.000) | (0.011) | (0.001) | (0.000) |
| $k_z^P$ | 0.002 | 0.000 | | 0.020 | 0.123 | 0.174 | 0.490 | | 1.558 | 6.635 | 0.535 | 3.235 |
| | (0.000) | (0.000) | | (0.023) | (0.084) | (0.035) | (0.038) | | (0.011) | (0.272) | (0.019) | (0.067) |
| $\theta_z^P$ | 0.000 | 0.000 | | 0.001 | 0.002 | 0.007 | 0.004 | | 0.003 | 0.002 | 0.001 | 0.001 |
| | (0.000) | (0.000) | | (0.001) | (0.001) | (0.001) | (0.000) | | (0.000) | (0.000) | (0.000) | (0.000) |
| $\beta$ | 0.084 | 0.000 | | 0.325 | 0.266 | 0.000 | 0.000 | | 0.000 | 0.000 | 0.000 | 0.071 |
| | (0.003) | (0.000) | | (0.002) | (0.012) | (0.000) | (0.000) | | (0.000) | (0.000) | (0.000) | (0.001) |
| $\tau_{cds}$ | 0.050 | 0.013 | | 0.205 | 0.236 | 0.322 | 0.412 | | 0.041 | 0.054 | 0.023 | 0.081 |
| | (0.001) | (0.000) | | (0.002) | (0.003) | (0.004) | (0.004) | | (0.001) | (0.000) | (0.001) | (0.002) |
| $\mu_j$ | 0.698 | 4.756 | | 0.236 | 0.165 | 0.097 | 0.069 | | 0.208 | 0.219 | 0.223 | 0.220 |
| | (0.004) | (0.001) | | (0.000) | (0.000) | (0.000) | (0.000) | | (0.000) | (0.001) | (0.001) | (0.001) |
| $v_j$ | 0.871 | 8.577 | | 0.000 | 0.000 | 0.009 | 0.000 | | 0.000 | 0.000 | 0.000 | 0.000 |
| | (0.004) | (0.005) | | (0.000) | (0.000) | (0.000) | (0.000) | | (0.000) | (0.000) | (0.000) | (0.000) |
| $\hat{\delta}_d^x$ | 0.231 | 0.374 | | 0.210 | 0.152 | 0.088 | 0.067 | | 0.187 | 0.197 | 0.200 | 0.198 |
| AIC | −36,242 | −35,502 | −73,603 | | | | | −84,851 | | | | |

Parameters $\beta$, which allow us to capture the correlation risk, are either some positive value or approximately zero. Therefore, it is possible that the intensity process and the stochastic volatility process do not have a common factor. Actually, in Section 4.3.3, we confirm that the impact of the stochastic volatility process is small in sovereign CDS spreads. The correlation parameter $\rho$ is negative for both periods. This result indicates that the

stochastic volatility process captures the negative slope of the volatility smile in currency markets as a leverage effect. The expected depreciation ratios, $\hat{\delta}_d^x$ $(= e^{-\mu_j + v_j/2} - 1)$, are 0.231 for Period 1 and 0.374 for Period 2. Although the range of the expected depreciation ratios, $\hat{\delta}_d^x$, is reasonable, both $\mu_j$ and $v_j$ present large values for Period 2. In Section 4.3.3, we observe that the impact of $\mu_j$ and $v_j$ is small and meaningless in the GBPUSD currency option markets for Period 2. Therefore, these two parameters are not identified.

Panel B in Table 2 shows the estimated parameters and their asymptotic standard errors for Eurozone countries. Although the risk-neutral mean reversion parameters $k_v^Q$ are positive, $k_{z_i}^Q$ are negative. Parameters $\beta$ are also either some positive value or approximately zero. Overall, these patterns are similar to the results for the UK. For Period 1, the expected depreciation $\hat{\delta}_d^x$ is smaller than approximately 0.2. The expected depreciation ratios in countries with a higher credit risk, such as Ireland and Portugal, is lower than that of countries with a lower credit risk, such as Spain and Italy. These results exhibit a trend similar to that found by Mano (2013). In contrast, in all countries, the expected depreciation ratio is approximately 0.2 for Period 2.

### 4.3.2. State Variables

Figure 1 shows the estimated state variables for the UK, and Figure 2 shows those for the Eurozone countries. For both periods, the intensity process is almost comprised an idiosyncratic risk process for all countries. Figures 1 and 2 show that the stochastic volatility process increases, as does the intensity process, in the UK and the Eurozone countries. However, the intensity process is similar to the idiosyncratic risk process, and, as a result, the stochastic volatility does not account for the intensity process. The risk-neutral behavior of both processes differ, as discussed in Section 4.3.1. Under physical measures, the result of the cross-correlation indicates that the stochastic volatility process and intensity process co-move; however, under the risk-neutral measure, the behaviors implied in the term structures differ.

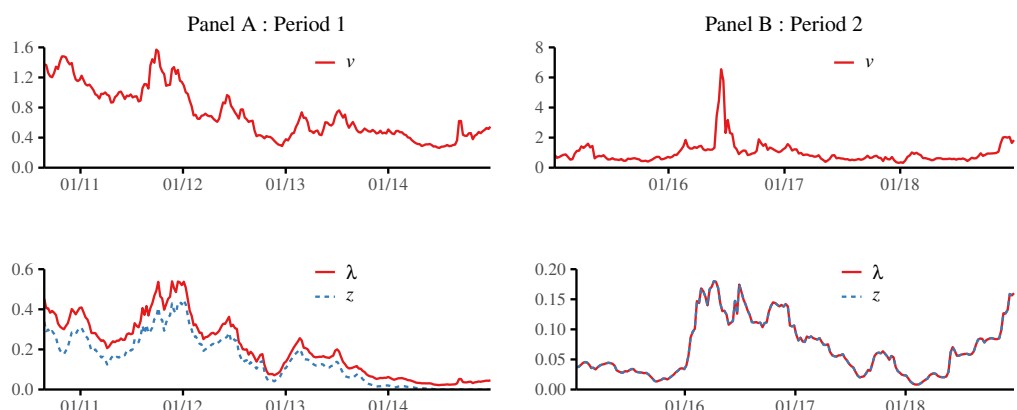

**Figure 1.** State variables and intensity processes estimated by the one-factor stochastic volatility model for the UK. The top figures show the time series of the stochastic volatility $v$. The bottom figures show the time series of the idiosyncratic risk process $z$ (blue dashed line) and the intensity process $\lambda$ (red line). In Panel A, Period 1 is from 25 August 2010 to 31 December 2014, with a weekly frequency. In Panel B, Period 2 is from 7 January 2015 to 26 December 2018, with a weekly frequency. The units are percentages.

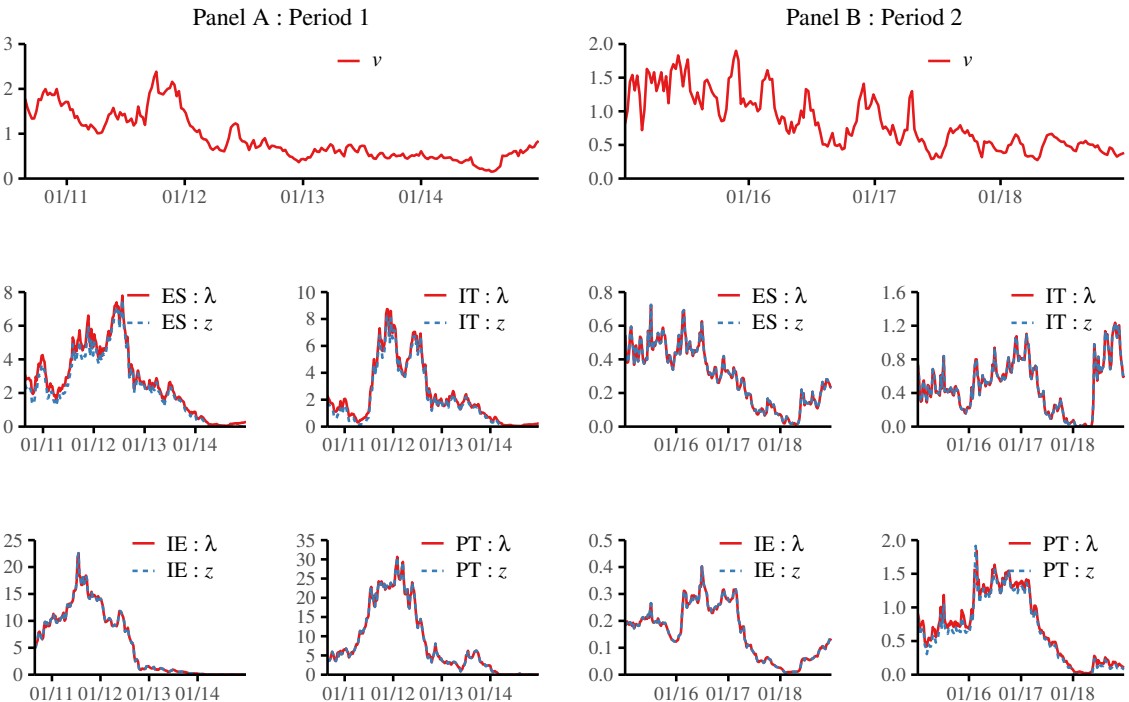

**Figure 2.** State variables and intensity processes estimated by the one-factor stochastic volatility model for the Eurozone countries. The top figures show the time series of stochastic volatility $v$. The middle and bottom figures show the time series of the idiosyncratic risk process $z$ (blue dashed line) and the intensity process $\lambda$ (red line) for Spain (ES), Italy (IT), Ireland (IE), and Portugal (PT). In Panel A, Period 1 is from 25 August 2010 to 31 December 2014 with a weekly frequency. In Panel B, Period 2 is from 7 January 2015 to 26 December 2018, with a weekly frequency. The units are percentages.

### 4.3.3. Decomposition

We measure the impact of each factor on each instrument to confirm their interactions. First, to measure the effect of the depreciation risk, we calculate the hypothetical foreign currency-denominated CDS spreads, $S^f_{\delta^d_x=0,t}$, without a depreciation risk. We assume that $\mu_j$ and $v_j$ are zero or that the depreciation ratio $\delta^d_x$ is zero and constant when calculating $S^f_{\delta^d_x=0,t}$. We derive the depreciation risk component as follows:

$$S^f_{\delta^d_x=0,t} - S^f_t.$$

We also derive the correlation risk component as follows:

$$S^d_t - S^f_{\delta^d_x=0,t}.$$

Additionally, to measure the impact of the co-movement between the intensity process and the stochastic volatility process, we calculate the hypothetical foreign currency-denominated CDS spreads, $S^f_{v_t=0,t}$, assuming that the stochastic volatility process $v_t$ is zero and constant. We derive the common component between the intensity process and stochastic volatility process as follows:

$$S^f_t - S^f_{v_t=0,t}.$$

Table 3 shows the summary statistics of the USD- and GBP/EUR-currency-denominated CDS spreads, the common component with the stochastic volatility, and the two quanto spread components at one- and five-year maturities. We find similar results in all panels.

On average, the GBP- and EUR-denominated CDS spreads are almost composed of the idiosyncratic risk process, and the common components with the stochastic volatility are small. Additionally, the quanto CDS spreads are mainly composed of the depreciation risk components. The standard deviation of the correlation risk component is also much smaller than that of the depreciation risk components. This result is consistent with the result that stochastic volatility does not account for the intensity process, as shown in Section 4.3.2.

**Table 3.** Decomposition of CDS spreads estimated by the one-factor stochastic volatility model for the UK and Eurozone countries. The Eurozone countries include Spain (ES), Italy (IT), Ireland (IE), and Portugal (PT). The table shows the decomposition of CDS spreads using the one-factor stochastic volatility model at one- and five-year maturities for each country. The quanto spreads, which are the differences between the USD- and GBP/EUR-denominated CDS spreads, are decomposed into a depreciation risk and a correlation risk component. The table shows the means and, in parentheses, standard deviations. The sample period of Period 1 is from 35 August 2010 to 31 December 2014, with a weekly frequency. The sample period of Period 2 is from 7 January 2015 to 26 December 2018, with a weekly frequency. The units are percentages.

| | | | | | | | | | USD CDS | |
| | | | GBP/EUR CDS | | Depreciation | | Correlation | | | |
| | Common ($v$) | | | | | | | | | |
| Maturity | 1Y | 5Y | 1Y | 5Y | 1Y | 5Y | 1Y | 5Y | 1Y | 5Y |
| *Panel A : UK* | | | | | | | | | | |
| Period1 | 0.03 | 0.04 | 0.13 | 0.37 | 0.04 | 0.11 | 0.00 | 0.00 | 0.17 | 0.49 |
| | (0.01) | (0.01) | (0.08) | (0.18) | (0.02) | (0.05) | (0.00) | (0.00) | (0.10) | (0.24) |
| Period2 | 0.00 | 0.00 | 0.04 | 0.15 | 0.02 | 0.09 | 0.00 | 0.00 | 0.07 | 0.23 |
| | (0.00) | (0.00) | (0.02) | (0.05) | (0.01) | (0.03) | (0.00) | (0.00) | (0.03) | (0.07) |
| *Panel B1 : ES* | | | | | | | | | | |
| Period1 | 0.13 | 0.12 | 1.45 | 2.13 | 0.39 | 0.52 | 0.00 | 0.00 | 1.84 | 2.65 |
| | (0.06) | (0.02) | (0.98) | (1.08) | (0.26) | (0.26) | (0.00) | (0.01) | (1.24) | (1.34) |
| Period2 | 0.00 | 0.00 | 0.20 | 0.54 | 0.05 | 0.12 | 0.00 | −0.01 | 0.25 | 0.65 |
| | (0.00) | (0.00) | (0.10) | (0.18) | (0.02) | (0.04) | (0.00) | (0.00) | (0.12) | (0.22) |
| *Panel B2 : IT* | | | | | | | | | | |
| Period1 | 0.12 | 0.11 | 1.43 | 2.20 | 0.26 | 0.35 | 0.00 | 0.00 | 1.68 | 2.55 |
| | (0.05) | (0.02) | (1.18) | (1.13) | (0.21) | (0.18) | (0.00) | (0.01) | (1.39) | (1.31) |
| Period2 | 0.00 | 0.00 | 0.38 | 0.95 | 0.09 | 0.18 | 0.00 | −0.01 | 0.47 | 1.12 |
| | (0.00) | (0.00) | (0.17) | (0.21) | (0.04) | (0.04) | (0.00) | (0.00) | (0.21) | (0.25) |
| *Panel B3 : IE* | | | | | | | | | | |
| Period1 | 0.00 | 0.00 | 3.30 | 3.35 | 0.32 | 0.28 | 0.00 | −0.01 | 3.61 | 3.62 |
| | (0.00) | (0.00) | (3.30) | (2.71) | (0.32) | (0.22) | (0.00) | (0.01) | (3.62) | (2.93) |
| Period2 | 0.00 | 0.00 | 0.11 | 0.33 | 0.03 | 0.08 | 0.00 | −0.01 | 0.13 | 0.41 |
| | (0.00) | (0.00) | (0.06) | (0.13) | (0.01) | (0.03) | (0.00) | (0.00) | (0.07) | (0.16) |
| *Panel B4 : PT* | | | | | | | | | | |
| Period1 | 0.00 | 0.00 | 5.16 | 5.32 | 0.36 | 0.29 | 0.00 | −0.01 | 5.52 | 5.60 |
| | (0.00) | (0.00) | (4.85) | (3.54) | (0.34) | (0.20) | (0.00) | (0.01) | (5.19) | (3.74) |
| Period2 | 0.03 | 0.02 | 0.51 | 1.41 | 0.12 | 0.18 | 0.00 | −0.01 | 0.63 | 1.58 |
| | (0.01) | (0.00) | (0.33) | (0.69) | (0.08) | (0.07) | (0.00) | (0.00) | (0.41) | (0.76) |

Next, to derive the depreciation component of the currency option implied volatility, we calculate the hypothetical currency option implied volatility, $IV_{\delta_x^d=0,t}$. We assume that $\mu_j$ and $v_j$ are equal to zero or that the depreciation ratio $\delta_x^d$ is zero and constant, when calculating $IV_{\delta_x^d=0,t}$. We derive the depreciation risk component as follows:

$$IV_t - IV_{\delta_x^d=0,t}.$$

Panel A in Table 4 provides the summary statistics of the implied volatility and depreciation risk components of the currency option at one-month and one-year maturities for GBPUSD. The depreciation risk effect is relatively small for the mean and standard deviation. The depreciation risk component does not constitute a large proportion of the implied volatility of the currency options. This finding is consistent with the result that the level of the intensity process is low and that a sovereign credit event is considered a rare event in the UK. Similarly, the standard deviations are small. As shown in Section 4.3.1, both $\mu_j$ and $v_j$ present large values. Because the depreciation risk components of the implied volatility are small, we cannot distinguish these parameters.

**Table 4.** Decomposition of the currency option implied volatility estimated by the one-factor stochastic volatility model. The table shows the decomposition of the currency option implied volatility using the one-factor stochastic volatility model at one-month and one-year maturities for each currency. The currency implied volatility is decomposed into a depreciation risk component. The table shows means and, in parentheses, standard deviations. The sample period of Period 1 is from 25 August 2010 to 31 December 2014, with a weekly frequency. The sample period of Period 2 is from 7 January 2015 to 26 December 2018, with a weekly frequency. The units are percentages.

| | Implied Volatility | | | | | | Depreciation | | | | | |
| | ATM | | 25-Delta Call | | 25-Delta Put | | ATM | | 25-Delta Call | | 25-Delta Put | |
| Maturity | 1M | 1Y | 1M | 1Y | 1M | 1Y | 1M | 1Y | 1M | 1Y | 1M | 1Y |
|---|---|---|---|---|---|---|---|---|---|---|---|---|
| *Panel A : GBPUSD* | | | | | | | | | | | | |
| Period1 | 8.33 | 8.82 | 8.20 | 8.41 | 8.50 | 9.49 | 0.05 | 0.20 | 0.05 | 0.22 | 0.07 | 0.30 |
| | (1.99) | (1.88) | (1.99) | (1.86) | (2.00) | (1.95) | (0.03) | (0.12) | (0.03) | (0.13) | (0.05) | (0.19) |
| Period2 | 9.45 | 9.47 | 8.99 | 8.83 | 9.99 | 10.42 | 0.03 | 0.16 | 0.03 | 0.18 | 0.04 | 0.22 |
| | (2.51) | (0.99) | (2.50) | (0.98) | (2.53) | (1.08) | (0.02) | (0.08) | (0.02) | (0.10) | (0.03) | (0.13) |
| *Panel B : EURUSD* | | | | | | | | | | | | |
| Period1 | 9.86 | 10.41 | 9.50 | 9.74 | 10.34 | 11.31 | 0.61 | 1.44 | 0.43 | 1.12 | 1.04 | 1.83 |
| | (2.85) | (2.66) | (2.78) | (2.50) | (3.23) | (2.84) | (0.51) | (1.00) | (0.36) | (0.81) | (0.88) | (1.20) |
| Period2 | 9.10 | 8.78 | 8.97 | 8.70 | 9.16 | 9.21 | 0.11 | 0.41 | 0.07 | 0.27 | 0.20 | 0.64 |
| | (1.97) | (1.27) | (2.01) | (1.24) | (2.05) | (1.31) | (0.06) | (0.19) | (0.04) | (0.12) | (0.11) | (0.30) |

Panels B1–B4 in Table 4 show the results for EURUSD. In Period 1 during the euro financial crisis, the depreciation risk components are up to 1.8% for the 25-delta put option, on average, which are higher than those for the at-the-money and 25-delta call options. Therefore, the depreciation risk component steepens the negative slope of the volatility smile. In Period 2, the euro expansion period, the depreciation risk effects are lower than those in Period 1.

## 5. Two-Factor Stochastic Volatility Model

In the one-factor stochastic volatility model, we find that the estimated intensity process does not include the stochastic volatility process, although the correlation values between the CDS spreads and currency option implied volatilities are positive under the physical measure. The estimated risk-neutral mean reversion parameters differ for these processes. This result captures the pattern in which the effects of the two processes on the slope of the term structure differ. In the one-factor stochastic volatility model, each process explains each market separately. Therefore, we introduce an idiosyncratic risk factor in stochastic volatility to capture the unique risk in the currency option market by keeping the first stochastic volatility process a common factor.

The SDE for the case where a single currency is only adopted by one country under a domestic risk-neutral measure, $\mathbb{Q}_d$, we can derive:

$$\frac{dX_t}{X_t} = (r_t^d - r_t^f)dt + \sqrt{v_{1,t}}dw_{1,x,t}^d + \sqrt{v_{2,t}}dw_{2,x,t}^d - (\delta_x^d dJ_t - \hat{\delta}_x^d \lambda_t^d dt),$$

$$dv_{1,t} = k_{v_1}^Q(\theta_{v_1}^Q - v_{1,t})dt + \sigma_{v_1}\sqrt{v_{1,t}}dw_{v_1,t}^d,$$

$$dv_{2,t} = k_{v_2}^Q(\theta_{v_2}^Q - v_{2,t})dt + \sigma_{v_2}\sqrt{v_{2,t}}dw_{v_2,t}^d,$$

$$\lambda_t^d = \beta v_{1,t} + z_t,$$

$$dz_t = k_z^Q(\theta_z^Q - z_t)dt + \sigma_z\sqrt{z_t}dw_t^z,$$

$$E[dw_{1,x,t}^d dw_{v_1,t}^d] = \rho_1 dt, \ E[dw_{2,x,t}^d dw_{v_2,t}^d] = \rho_2 dt,$$

$$E[dw_{v_1,t}^d dw_{v_2,t}^d] = 0, \ E[dw_{1,x,t}^d dw_t^z] = 0, \ E[dw_{2,x,t}^d dw_t^z] = 0.$$

The added process $v_{2,t}$ is expected to capture the idiosyncratic risk in the currency market. Furthermore, we can derive the diffusion process under the foreign risk-neutral measure $\mathbb{Q}_f$ as:

$$
\begin{aligned}
\lambda_t^f &= (1 - \hat{\delta}_x^d)\lambda_t^d, \\
dv_{1,t} &= (k_{v_1}^Q \theta_{v_1}^Q - (k_{v_1}^Q - \rho_1 \sigma_{v_1})v_{1,t})dt + \sigma_{v_1}\sqrt{v_{1,t}}dw_{v_1,t}^f, \\
dz_t &= k_z^Q(\theta_z^Q - z_t)dt + \sigma_z\sqrt{z_t}dw_t^z.
\end{aligned}
$$

Next, we describe the model for the case wherein multiple countries adopt the same currency as a currency union. The SDE under the domestic risk-neutral measure $\mathbb{Q}_d$ can be derived as follows:

$$
\begin{aligned}
\frac{dX_t}{X_t} &= (r_t^d - r_t^f)dt + \sqrt{v_{1,t}}dw_{1,x,t}^d + \sqrt{v_{2,t}}dw_{2,x,t}^d - \sum_{i=1}^N (\delta_{x,i}^d dJ_{I,t} - \hat{\delta}_{x,i}^d \lambda_{i,t}^d dt), \\
dv_{1,t} &= k_{v_1}^Q(\theta_{v_1}^Q - v_{1,t})dt + \sigma_{v_1}\sqrt{v_{1,t}}dw_{v_1,t}^d, \\
dv_{2,t} &= k_{v_2}^Q(\theta_{v_2}^Q - v_{2,t})dt + \sigma_{v_2}\sqrt{v_{2,t}}dw_{v_2,t}^d, \\
\lambda_{i,t}^d &= \beta_i v_{1,t} + z_{i,t}, \\
dz_{i,t} &= k_{z_i}^Q(\theta_{z_i}^Q - z_{i,t})dt + \sigma_{z_i}\sqrt{z_{i,t}}dw_{i,t}^z, \\
&\quad E[dw_{1,x,t}^d dw_{v_1,t}^d] = \rho_1 dt, \ E[dw_{2,x,t}^d dw_{v_2,t}^d] = \rho_2 dt, \\
&\quad E[dw_{v_1,t}^d dw_{v_2,t}^d] = 0, \ E[dw_{1,x,t}^d dw_{i,t}^z] = 0, \ E[dw_{2,x,t}^d dw_{i,t}^z] = 0.
\end{aligned}
$$

Furthermore, we derive the diffusion process under the foreign risk-neutral measure $\mathbb{Q}_f$, as follows:

$$
\begin{aligned}
\lambda_{i,t}^f &= (1 - \hat{\delta}_{x,i}^d)\lambda_{i,t}^d, \\
dv_{1,t} &= (k_{v_1}^Q \theta_{v_1}^Q - (k_{v_1}^Q - \rho_1 \sigma_{v_1})v_{1,t})dt + \sigma_{v_1}\sqrt{v_{1,t}}dw_{v_1,t}^f, \\
dz_{i,t} &= k_{z_i}^Q(\theta_{z_i}^Q - z_{i,t})dt + \sigma_{z_i}\sqrt{z_{i,t}}dw_t^{z_i}.
\end{aligned}
$$

To estimate the two stochastic volatility processes, we assume that they follow the square-root process under the physical measure, as follows:

$$
\begin{aligned}
dv_{1,t} &= k_{v_1}^P(\theta_{v_1}^P - v_{1,t})dt + \sigma_v\sqrt{v_{1,t}}dw_{v_1,t}^P, \\
dv_{2,t} &= k_{v_2}^P(\theta_{v_2}^P - v_{2,t})dt + \sigma_v\sqrt{v_{2,t}}dw_{v_2,t}^P.
\end{aligned}
$$

For the stochastic volatility processes $v_{1,t}$ and $v_{2,t}$, the market price of risk is assumed to follow a formula similar to the idiosyncratic risk process $z_{i,t}$, as detailed in Section 4.2.

### 5.1. Estimated Parameters

Panel A in Table 5 provides the estimated parameters and their asymptotic standard errors for the UK. The levels of parameter $\beta$ are approximately zero. These results are the same as those of the one-factor stochastic volatility model, and the idiosyncratic risk process are almost the same as those of the one-factor stochastic model. The standard deviations of the observation error for currency options in the two-factor stochastic volatility model are lower than those in the one-factor stochastic volatility model. Therefore, we add a second factor to the stochastic volatility to improve the fit of the model to the currency option for both periods. Additionally, the Akaike information criterion (AIC) in the two-factor stochastic volatility model are lower than those in the one-factor stochastic volatility model. The expected depreciation ratio, $\hat{\delta}_d^x$, is almost the same as that of the two-factor stochastic volatility model.

**Table 5.** Estimated parameters from the two-factor stochastic volatility model for the UK and the Eurozone countries. The Eurozone countries are Spain (ES), Italy (IT), Ireland (IE), and Portugal (PT). The table shows the quasi-maximum likelihood estimator. The asymptotic standard errors are in parentheses. The asymptotic standard errors are based on the inverse of the information matrix computed from the Hessian matrix and gradient vector for the log-likelihood function. In Panel A, Period 1 is from 25 August 2010 to 31 December 2014, with a weekly frequency. In Panel B, Period 2 is from 7 January 2015 to 26 December 2018, with a weekly frequency.

| | Panel A: UK | | | Panel B: Eurozone Countries | | | | | | | |
|---|---|---|---|---|---|---|---|---|---|---|---|
| | Period 1 | Period 2 | | Period 1 | | | | Period 2 | | | |
| | | | | Country | | | | Country | | | |
| | | | | ES | IT | IE | PT | ES | IT | IE | PT |
| $k_{v_1}^Q$ | 3.802 (0.000) | −0.084 (0.037) | 0.002 (0.004) | | | | 2.052 (0.010) | | | | |
| $k_{v_1}^Q \theta_{v_1}^Q$ | 0.024 (0.029) | 0.001 (0.021) | 0.001 (0.001) | | | | 0.008 (0.003) | | | | |
| $\sigma_{v_1}$ | 0.453 (0.007) | 0.111 (0.011) | 0.098 (0.001) | | | | 0.474 (0.013) | | | | |
| $k_{v_1}^P$ | 2.571 (0.029) | 0.000 (0.000) | 1.835 (0.476) | | | | 18.051 (1.104) | | | | |
| $k_{v_1}^P \theta_{v_1}^P$ | 0.022 (0.000) | 0.000 (0.000) | 0.000 (0.000) | | | | 0.003 (0.000) | | | | |
| $\rho_1$ | −0.243 (0.004) | −0.305 (0.012) | −1.000 (0.000) | | | | −0.489 (0.003) | | | | |
| $k_{v_2}^Q$ | −0.136 (0.004) | 10.897 (0.252) | 0.689 (0.023) | | | | 0.062 (0.002) | | | | |
| $k_{v_2}^Q \theta_{v_2}^Q$ | 0.001 (0.002) | 0.048 (0.689) | 0.005 (0.016) | | | | 0.000 (0.000) | | | | |
| $\sigma_{v_2}$ | 0.160 (0.002) | 0.650 (0.011) | 0.065 (0.003) | | | | 0.040 (0.000) | | | | |
| $k_{v_2}^P$ | 0.000 (0.000) | 0.079 (0.072) | 0.650 (0.037) | | | | 2.081 (0.028) | | | | |
| $\theta_{v_2}^P$ | 0.000 (0.000) | 0.000 (0.000) | 0.010 (0.000) | | | | 0.005 (0.000) | | | | |
| $\rho_2$ | −0.465 (0.009) | −0.743 (0.006) | −0.313 (0.063) | | | | 0.483 (0.003) | | | | |
| $\tau_{cur}$ | 0.264 (0.006) | 0.411 (0.033) | 0.542 (0.038) | | | | 0.285 (0.004) | | | | |
| $k_z^Q$ | −0.344 (0.002) | −0.360 (0.001) | | −0.232 (0.001) | −0.524 (0.011) | 0.049 (0.005) | −0.141 (0.014) | −0.284 (0.002) | −0.259 (0.015) | −0.354 (0.010) | −0.523 (0.003) |
| $k_z^Q \theta_z^Q$ | 0.001 (0.001) | 0.000 (0.000) | | 0.000 (0.000) | 0.000 (0.000) | 0.001 (0.003) | 0.004 (0.003) | 0.001 (0.001) | 0.004 (0.003) | 0.001 (0.005) | 0.002 (0.002) |
| $\sigma_z$ | 0.040 (0.000) | 0.029 (0.001) | | 0.162 (0.000) | 0.262 (0.002) | 0.198 (0.004) | 0.423 (0.009) | 0.074 (0.000) | 0.264 (0.012) | 0.030 (0.001) | 0.242 (0.001) |
| $k_z^P$ | 0.000 (0.000) | 0.000 (0.000) | | 0.688 (0.001) | 0.003 (0.000) | 0.001 (0.000) | 0.006 (0.007) | 1.497 (0.007) | 7.118 (0.274) | 0.543 (0.018) | 2.619 (0.086) |
| $\theta_z^P$ | 0.000 (0.000) | 0.000 (0.000) | | 0.003 (0.001) | 0.000 (0.000) | 0.000 (0.000) | 0.000 (0.000) | 0.003 (0.000) | 0.002 (0.000) | 0.001 (0.000) | 0.001 (0.000) |
| $\beta$ | 0.000 (0.000) | 0.021 (0.001) | | 11.864 (0.036) | 16.883 (0.085) | 10.310 (0.050) | 26.080 (0.016) | 0.000 (0.000) | 0.000 (0.000) | 0.000 (0.000) | 0.084 (0.001) |
| $\tau_{cds}$ | 0.050 (0.001) | 0.013 (0.000) | | 0.148 (0.001) | 0.107 (0.000) | 0.326 (0.005) | 0.389 (0.006) | 0.041 (0.001) | 0.053 (0.000) | 0.023 (0.001) | 0.082 (0.002) |
| $\mu_j$ | 0.291 (0.003) | 0.463 (0.001) | | 0.206 (0.006) | 0.133 (0.003) | 0.084 (0.001) | 0.045 (0.003) | 0.204 (0.000) | 0.226 (0.002) | 0.221 (0.001) | 0.174 (0.001) |
| $v_j$ | 0.000 (0.000) | 0.000 (0.000) | | 0.000 (0.000) | 0.000 (0.000) | 0.020 (0.000) | 0.000 (0.000) | 0.000 (0.000) | 0.000 (0.000) | 0.000 (0.000) | 0.000 (0.000) |
| $\hat{\delta}_d^x$ | 0.252 | 0.371 | | 0.186 | 0.124 | 0.072 | 0.044 | 0.185 | 0.202 | 0.198 | 0.160 |
| $AIC$ | −38,602 | −37,272 | −76,010 | | | | −87,775 | | | | |

Next, we consider the Eurozone market. Panel B in Table 5 shows the estimated parameters and their asymptotic standard errors for the Eurozone countries. First, we focus on the results for Period 1. Except for Ireland's CDS spreads, the standard deviations of the observation error for both instruments in the two-factor stochastic volatility model are lower than those in the one-factor stochastic volatility model. The standard deviation of the observation error for Ireland's CDS spreads is almost the same as that of the one-factor stochastic volatility model. The levels of parameter $\beta_i$ are much higher than those in the one-factor stochastic volatility model, indicating that the intensity processes are composed of the first stochastic volatility process. We confirm that the effect of the first stochastic volatility process is the common risk factor in the intensity process in Section 5.2. The

risk-neutral mean reversion parameter $k_{v_2}^Q$ is positive and similar to $k_v^Q$ in the one-factor stochastic volatility model. On the contrary, the risk-neutral mean reversion parameters $k_{v_1}^Q$ are approximately zero and have a value between the mean reversion parameters of the stochastic volatility and idiosyncratic risk in the one-factor stochastic volatility model. By contrast, the risk-neutral mean reversion parameters $k_{z_i}^Q$ are lower than those in the one-factor stochastic volatility model, except for Ireland. $k_{v_1}^Q$ and $k_{z_i}^Q$ imply that the stochastic volatility $v_{1,t}$ and idiosyncratic risk $z_{i,t}$ are not stationary under the risk-neutral measure. Additionally, the risk-neutral mean reversion parameters $k_{v_1}^Q$ and $k_{z_i}^Q$ are lower than $k_{v_1}^P$ and $k_{z_i}^P$ under the physical measure except for Ireland's $k_{z_i}^Q$. These differences between the probability measures are indicative of risk premiums. Pan and Singleton (2008) and Carr and Wu (2007) show the same tendency in the intensity process of sovereign credit risk. We will discuss the risk premiums related to these results in more detail in Section 5.5.

The correlation parameter of the stochastic volatility $v_{1,t}$, which is a part of the intensity process, is approximately one. This level is larger than that of the stochastic volatility $v_{2,t}$.

Christoffersen et al. (2009) present the correlation formula in the two-factor stochastic volatility model between the underlying asset and its conditional variance as follows:

$$
\begin{aligned}
Cor_t[dX/X, dV] &= \frac{\sigma_{v_1}\rho_1 v_1 + \sigma_{v_2}\rho_2 v_2}{\sqrt{\sigma_{v_1}^2 v_1 + \sigma_{v_2}^2 v_2}\sqrt{v_1 + v_2}}, \\
&= \frac{v_r(\sigma_{v_1}\rho_1 - \sigma_{v_2}\rho_2) + \sigma_{v_2}\rho_2}{\sqrt{v_r(\sigma_{v_1}^2 - \sigma_{v_2}^2) + \sigma_{v_2}^2}},
\end{aligned}
$$

where $Var_t[dX/X] = (v_1 + v_2)dt := Vdt$ and $v_r = v_1/(v_1 + v_2)$. Therefore, the actual value of the correlation between the exchange rate and its conditional variance is stochastic in time and depends on the share of each stochastic volatility process in total volatility. Consequently, although $\rho_1$ equals one for Period 1, the actual correlations are not necessarily equal to one in the two-factor stochastic volatility model.

The expected depreciation ratio $\hat{\delta}_d^x$ is lower than that of the one-factor stochastic volatility models by approximately 2%. Additionally, the AIC values in the two-factor stochastic volatility model are lower than those in the one-factor stochastic volatility model.

Next, we focus on the results for Period 2. The pattern is almost the same as that for the UK. The standard deviation of the observation error for the currency options in the two-factor stochastic volatility model is lower than that in the one-factor stochastic volatility model. The levels of $\beta$ are not high and are the same as those in the one-factor stochastic volatility model. The expected depreciation ratio $\hat{\delta}_d^x$ is almost the same as that of the two-factor stochastic volatility model.

In Figures 3–5, we show the model implied CDS spreads and currency options implied volatility with the observed data. We can see that the overall fit is good.

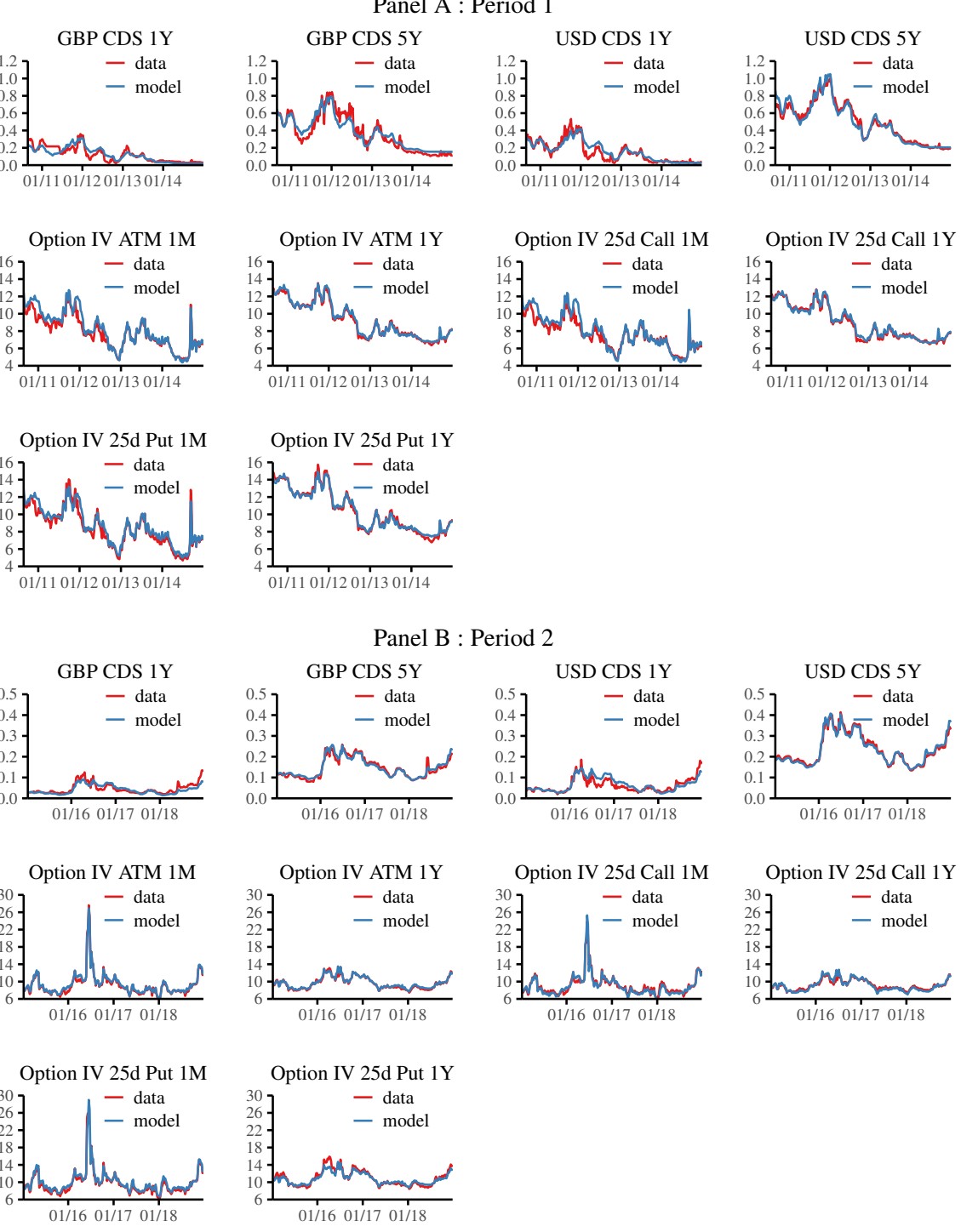

**Figure 3.** CDS spreads and currency options implied volatility data, as well as model implied theoretical CDS spreads and currency options implied volatility estimated by the two-factor stochastic volatility model for the UK, for the GBPUSD. The top figures show the observed GBP- and USD-denominated CDS spreads data (red line) and model implied theoretical GBP- and USD-denominated CDS spreads (blue line) at one- and five-year maturities. The middle and bottom figures show the observed currency option implied volatilities data (red line) and model implied theoretical currency option implied volatilities (blue line) of the at-the-money (ATM), 25-delta call, and 25-delta put at one-month and one-year maturities. In Panel A, Period 1 is from 25 August 2010 to 31 December 2014, with a weekly frequency. In Panel B, Period 2 is from 7 January 2015 to 26 December 2018, with a weekly frequency. The units are percentages.

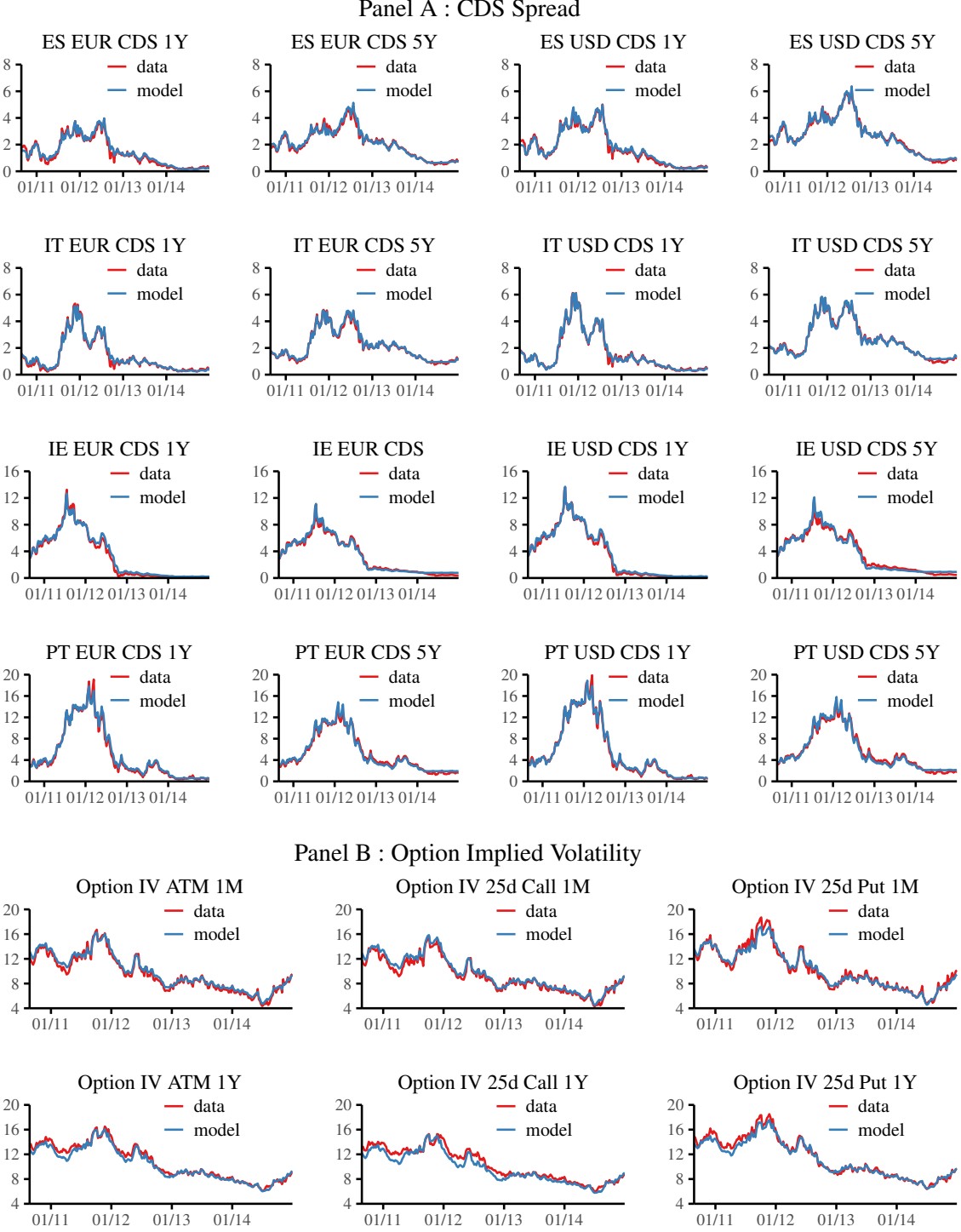

**Figure 4.** CDS spreads and currency options implied volatility data, as well as model implied theoretical CDS spreads and currency options implied volatility estimated by the two-factor stochastic volatility model for the Eurozone countries and for the EURUSD in Period 1. The figures in Panel A show the observed EUR- and USD-denominated CDS spreads data (red line) and model implied theoretical EUR- and USD-denominated CDS spreads (blue line) at one- and five-year maturities for Spain (ES), Italy (IT), Ireland (IE), and Portugal (PT). The bottom figures show the observed currency option implied volatilities data (red line) and model implied theoretical currency option implied volatilities (blue line) of the at-the-money (ATM), 25-delta call, and 25-delta put at one-month and one-year maturities. The sample period is from 25 August 2010 to 31 December 2014, with a weekly frequency. The units are percentages.

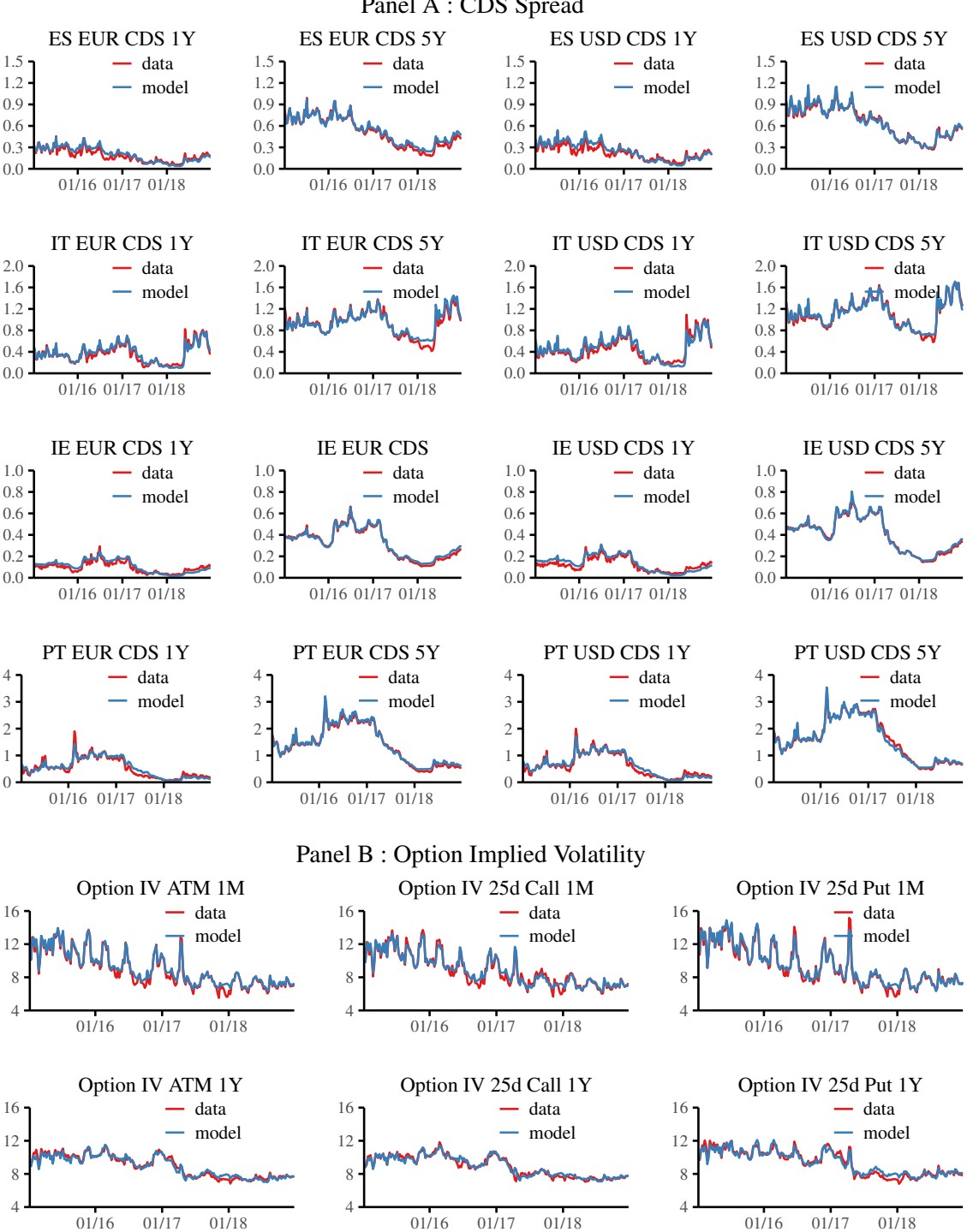

**Figure 5.** CDS spreads and currency options implied volatility data, as well as model implied theoretical CDS spreads and currency options implied volatility estimated by the two-factor stochastic volatility model for the Eurozone countries and for the EUR in Period 2. The figures in Panel A show the observed EUR- and USD-denominated CDS spreads data (red line) and model implied theoretical EUR- and USD-denominated CDS spreads (blue line) at one- and five-year maturities for Spain (ES), Italy (IT), Ireland (IE), and Portugal (PT). The bottom figures show the observed currency option implied volatilities data (red line) and model implied theoretical currency option implied volatilities (blue line) of the at-the-money (ATM), 25-delta call, and 25-delta put at one-month and one-year maturities. The sample period is from 7 January 2015 to 26 December 2018, with a weekly frequency. The units are percentages.

### 5.2. State Variables

Figure 6 shows the state variables estimated using a two-factor stochastic volatility model for the UK. The intensity process is similar to the idiosyncratic risk process, and the stochastic volatility process $v_{1,t}$ does not account for the intensity process, which is the same result as in the one-factor stochastic volatility model.

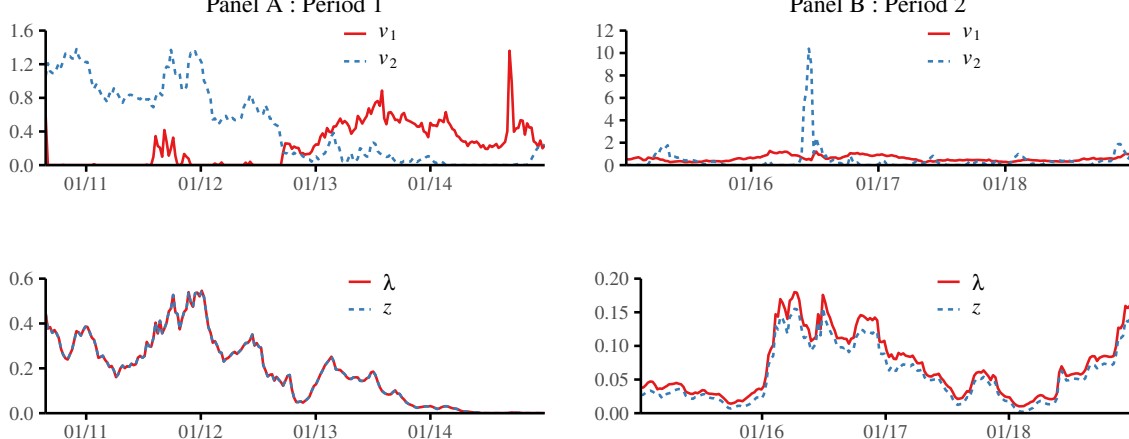

**Figure 6.** State variables and intensity processes estimated by the two-factor stochastic volatility model for the UK. The top figures show the time series of the stochastic volatilities $v_1$ (red line) and $v_2$ (blue dashed line). The bottom figures show the time series of the idiosyncratic risk process $z$ (blue dashed line) and intensity process $\lambda$ (red line). In Panel A, Period 1 is from 25 August 2010 to 31 December 2014, with a weekly frequency. In Panel B, Period 2 is from 7 January 2015 to 26 December 2018, with a weekly frequency. The units are percentages.

Figure 7 provides the estimated hidden process for Eurozone countries. Panel A in Figure 7 provides the results for Period 1 during the Eurozone's sovereign debt crisis. The stochastic volatility $v_{1,t}$ increased from July 2011. Portugal and Greece were downgraded in July 2011. Moreover, these fears triggered an increase in the CDS spreads for Italy and Spain. In contrast, as the CDS spreads decreased in Italy and Spain, the stochastic volatility $v_{1,t}$ decreased. Thus, the intensity process is higher than the idiosyncratic risk process, and the stochastic volatility $v_{1,t}$ accounts for a large part of the intensity process during periods of high market stress, especially in Italy and Spain. As discussed in Section 5.1, the risk-neutral mean reversion parameter $k_{v_1}^Q$ represents the value between the mean reversion parameters of the stochastic volatility and idiosyncratic risk in the one-factor stochastic volatility model. During periods of high market stress, the mean reversion of the intensity process is indicated to become higher in the sovereign CDS market, and that of the stochastic volatility becomes lower in the currency market under risk-neutral measures. This effect is captured by the stochastic volatility process $v_{1,t}$ in the two-factor model.

Panel B in Figure 7 provides the results for Period 2 during the Eurozone expansion period. The relationship between the intensity and stochastic volatility process is similar to that of the one-factor stochastic volatility model.

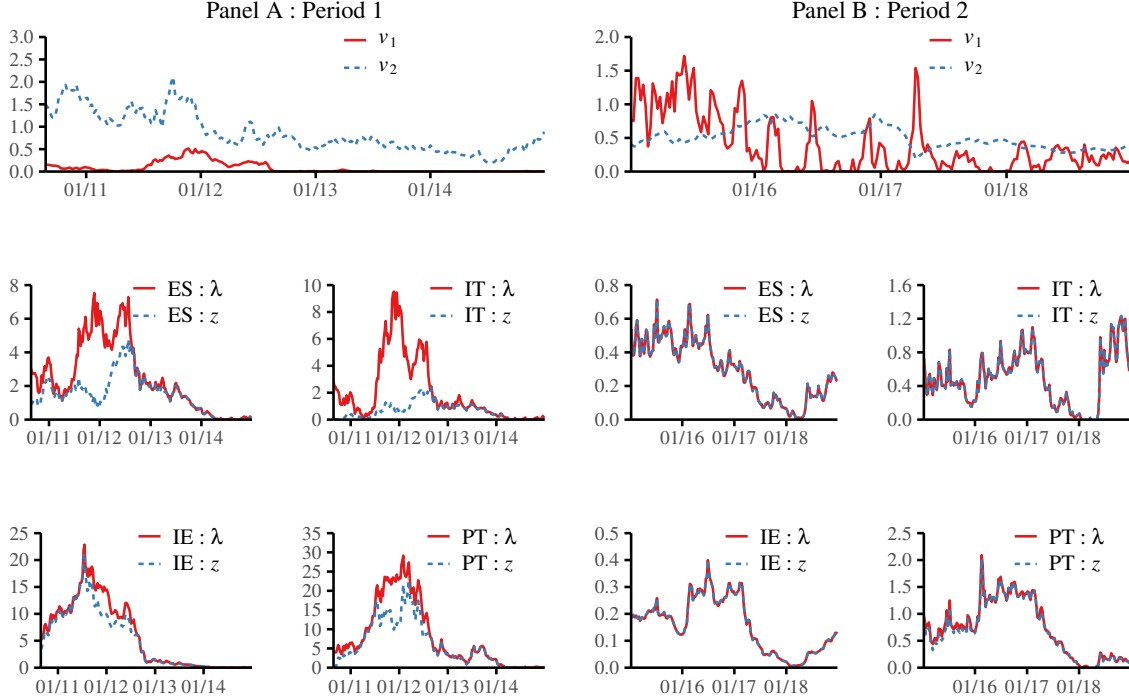

**Figure 7.** State variables estimated by the two-factor stochastic volatility model for the Eurozone countries. The top figures show the time series of the stochastic volatilities $v_1$ (red line) and $v_2$ (blue dashed line). The middle and bottom figures show the time series of the idiosyncratic risk process $z_t$ (blue dashed line) and intensity process $\lambda$ (red line) for Spain (ES), Italy (IT), Ireland (IE), and Portugal (PT). In Panel A, Period 1 is from 25 August 2010 to 31 December 2014, with a weekly frequency. In Panel B, Period 2 is from 7 January 2015 to 26 December 2018, with a weekly frequency. The units are percentages.

*5.3. Model Comparison*

We discuss the performance and test the statistical significance of different stochastic volatility models in the sovereign CDS spreads and currency options. We calculate the theoretical CDS spreads and currency option implied volatility based on the estimated parameters and state variables for both models. We use the likelihood ratio test proposed by Vuong (1989) for non-nested models. As described in Section 4.2, although the conditional distribution of the intensity and stochastic volatility processes is a non-central, Chi-square distribution that is not Gaussian, we treat these processes as if the state variables were conditionally normally distributed. Although the innovations are not normally distributed, we use the likelihood ratio test as conducted by Chen and Scott (2003).[14] In currency option implied volatility, the effect of depreciation risk is higher than that of the stochastic volatility process $v_{1,t}$. For the euro expansion period, the depreciation risk effects are lower than those in Period 1. We thus define $LR(\Theta_i, \Theta_j)$ as the difference in the log-likelihood $L(\Theta)$ between models $i$ and $j$:

$$LR(\Theta_i, \Theta_j) = L(\Theta_i) - L(\Theta_j).$$

We calculate the following test statistic:

$$M = \frac{LR(\Theta_i, \Theta_j)}{\sqrt{ns}},$$

where $n$ is the number of time series and $s^2$ is the variance of the difference between the log-likelihoods $l_{i,t}$ and $l_{j,t}$. $l_{i,t}$ is the log-likelihood of the model $i$ at a time $t$. Following Vuong (1989), $M$ has a normal distribution with zero mean and an asymptotic unit variance.

We estimate $s$ from the difference between the log-likelihoods $(l_{i,t} - l_{j,t})$, which is adjusted for heteroskedasticity and autocorrelation following Newey and West (1987), with the optimal number of lags according to Andrews (1991). The test statistic $M$ for the results for the UK is 21.59 for Period 1 and 17.17 for Period 2. For the Eurozone countries, it is 8.03 for Period 1 and 6.97 for Period 2. These statistics are higher than 1.65, corresponding to a 95% one-sided confidence level. These results indicate that the two-factor stochastic volatility model performs significantly better than the one-factor stochastic volatility model.

*5.4. Decomposition*

Panel A in Table 6 provides summary statistics for the USD- and GBP-denominated CDS spreads, the common component with the stochastic volatility process, and the two components of the quanto spreads at one- and five-year maturities for the UK. The patterns of the decomposed components are identical for both periods. As a common risk factor, the stochastic volatility process $v_{1,t}$ accounts for a small part of the intensity process. The quanto spreads are mainly composed of depreciation risk effects, on average. The standard deviations of the common and correlation risk components are also small. These results are similar to those of the one-factor stochastic volatility model.

**Table 6.** Decomposition of CDS spreads estimated by the two-factor stochastic volatility model for the UK and the Eurozone countries. The Eurozone countries are Spain (ES), Italy (IT), Ireland (IE), and Portugal (PT). The table shows the decomposition of CDS spreads using the two-factor stochastic volatility model at one- and five-year maturities for each country. The quanto spreads, which are the differences between the USD- and GBP/EUR-denominated CDS spreads, are decomposed into a depreciation risk and a correlation risk component. The table shows the means and, in parentheses, standard deviations. The sample period of Period 1 is from 25 August 2010 to 31 December 2014, with a weekly frequency. The sample period of Period 2 is from 7 January 2015 to 26 December 2018, with a weekly frequency. The units are percentages.

| | | | | | | | | | USD CDS | |
| | | | GBP/EUR CDS | | Depreciation | | Correlation | | | |
| | Common($v$) | | | | | | | | | |
| Maturity | 1Y | 5Y | 1Y | 5Y | 1Y | 5Y | 1Y | 5Y | 1Y | 5Y |
|---|---|---|---|---|---|---|---|---|---|---|
| *Panel A : UK* | | | | | | | | | | |
| Period1 | 0.00 | 0.00 | 0.12 | 0.37 | 0.04 | 0.12 | 0.00 | 0.00 | 0.16 | 0.50 |
| | (0.00) | (0.00) | (0.08) | (0.18) | (0.03) | (0.06) | (0.00) | (0.00) | (0.11) | (0.24) |
| Period2 | 0.01 | 0.01 | 0.04 | 0.15 | 0.02 | 0.09 | 0.00 | 0.00 | 0.07 | 0.23 |
| | (0.00) | (0.00) | (0.02) | (0.05) | (0.01) | (0.03) | (0.00) | (0.00) | (0.03) | (0.07) |
| *Panel B1 : ES* | | | | | | | | | | |
| Period1 | 0.61 | 0.93 | 1.44 | 2.11 | 0.33 | 0.41 | 0.03 | 0.16 | 1.80 | 2.67 |
| | (0.67) | (0.45) | (1.02) | (1.08) | (0.23) | (0.20) | (0.04) | (0.09) | (1.29) | (1.34) |
| Period2 | 0.00 | 0.00 | 0.20 | 0.54 | 0.05 | 0.12 | 0.00 | −0.01 | 0.25 | 0.65 |
| | (0.00) | (0.00) | (0.10) | (0.18) | (0.02) | (0.04) | (0.00) | (0.00) | (0.12) | (0.22) |
| *Panel B2 : IT* | | | | | | | | | | |
| Period1 | 0.93 | 1.33 | 1.38 | 2.17 | 0.19 | 0.21 | 0.05 | 0.19 | 1.62 | 2.57 |
| | (1.02) | (0.63) | (1.21) | (1.13) | (0.17) | (0.10) | (0.06) | (0.11) | (1.44) | (1.31) |
| Period2 | 0.00 | 0.00 | 0.37 | 0.95 | 0.09 | 0.18 | 0.00 | −0.01 | 0.47 | 1.12 |
| | (0.00) | (0.00) | (0.17) | (0.21) | (0.04) | (0.04) | (0.00) | (0.00) | (0.21) | (0.25) |
| *Panel B3 : IE* | | | | | | | | | | |
| Period1 | 0.61 | 0.94 | 3.35 | 3.30 | 0.26 | 0.23 | 0.03 | 0.14 | 3.64 | 3.67 |
| | (0.68) | (0.47) | (3.33) | (2.69) | (0.26) | (0.19) | (0.04) | (0.09) | (3.61) | (2.94) |
| Period2 | 0.00 | 0.00 | 0.11 | 0.33 | 0.03 | 0.08 | 0.00 | −0.01 | 0.13 | 0.41 |
| | (0.00) | (0.00) | (0.06) | (0.13) | (0.01) | (0.03) | (0.00) | (0.00) | (0.07) | (0.16) |
| *Panel B4 : PT* | | | | | | | | | | |
| Period1 | 1.57 | 2.09 | 5.20 | 5.23 | 0.23 | 0.17 | 0.07 | 0.23 | 5.50 | 5.64 |
| | (1.74) | (1.06) | (4.85) | (3.51) | (0.22) | (0.12) | (0.09) | (0.13) | (5.14) | (3.74) |
| Period2 | 0.02 | 0.02 | 0.53 | 1.43 | 0.10 | 0.16 | 0.00 | −0.01 | 0.63 | 1.58 |
| | (0.01) | (0.00) | (0.34) | (0.69) | (0.06) | (0.07) | (0.00) | (0.00) | (0.41) | (0.76) |

Panel A in Table 7 provides the summary statistics of the implied volatility and its depreciation component of the currency option at one-month and one-year maturities for GBPUSD. For the average and standard deviation, the depreciation risk effect is small for the GBPUSD option.

**Table 7.** Decomposition of currency option implied volatility estimated by the two-factor stochastic volatility model. The table shows the decomposition of the currency option implied volatility using the two-factor stochastic volatility model at one-month and one-year maturities for each currency. The currency implied volatility is decomposed into a depreciation risk component. The table shows the means and, in parentheses, standard deviations. The sample period of Period 1 is from 25 August 2010 to 31 December 2014, with a weekly frequency. The sample period of Period 2 is from 7 January 2015 to 26 December 2018, with a weekly frequency. ATM: at-the-money. The units are percentages.

| | Implied Volatility | | | | | | Depreciation | | | | | |
| | ATM | | 25-Delta Call | | 25-Delta Put | | ATM | | 25-Delta Call | | 25-Delta Put | |
| Maturity | 1M | 1Y | 1M | 1Y | 1M | 1Y | 1M | 1Y | 1M | 1Y | 1M | 1Y |
|---|---|---|---|---|---|---|---|---|---|---|---|---|
| *Panel A : GBPUSD* | | | | | | | | | | | | |
| Period1 | 8.16 | 9.32 | 7.95 | 8.96 | 8.69 | 10.43 | 0.02 | 0.07 | 0.01 | 0.04 | 0.03 | 0.10 |
| | (2.16) | (2.09) | (2.07) | (1.88) | (2.08) | (2.30) | (0.01) | (0.04) | (0.01) | (0.03) | (0.02) | (0.06) |
| Period2 | 9.42 | 9.74 | 8.96 | 9.12 | 10.07 | 10.77 | 0.01 | 0.04 | 0.01 | 0.03 | 0.02 | 0.07 |
| | (2.66) | (1.31) | (2.50) | (1.34) | (2.89) | (1.32) | (0.01) | (0.02) | (0.00) | (0.01) | (0.01) | (0.03) |
| *Panel B : EURUSD* | | | | | | | | | | | | |
| Period1 | 9.85 | 10.36 | 9.60 | 9.80 | 10.27 | 11.22 | 0.49 | 1.12 | 0.37 | 0.89 | 0.81 | 1.48 |
| | (2.86) | (2.63) | (2.77) | (2.43) | (3.17) | (2.93) | (0.44) | (0.83) | (0.33) | (0.68) | (0.73) | (1.03) |
| Period2 | 9.09 | 8.91 | 8.90 | 8.92 | 9.44 | 9.42 | 0.10 | 0.35 | 0.07 | 0.23 | 0.18 | 0.56 |
| | (1.99) | (1.24) | (1.87) | (1.19) | (2.28) | (1.36) | (0.05) | (0.15) | (0.03) | (0.10) | (0.10) | (0.23) |

Next, we consider the Eurozone market. Panels B1–B4 in Table 6 provide the summary statistics of the USD- and EUR-denominated CDS spreads, the common component with the stochastic volatility process, and two components of the quanto spreads at one- and five-year maturities for the Eurozone countries. In Period 1, during the European sovereign crisis, the stochastic volatility $v_{1,t}$ accounts for approximately half of the EUR-denominated CDS spreads on average, especially in Spain and Italy. This finding is consistent with the result that stochastic volatility accounts for the intensity process. The averages of the correlation risk component are higher than those for other cases, especially at the five-year maturity. The correlation risk components are either the same or less than the depreciation risk components at the five-year maturity. Overall, the depreciation risk is the main driver of quanto CDS spreads. The two-factor stochastic volatility model captures the co-movement and correlation risk in the CDS spreads. These findings are consistent with the result that the expected depreciation risk values are lower than those of the one-factor stochastic volatility model.

Period 2 includes the European expansion period. These results are almost identical to those of the one-factor stochastic volatility model. Stochastic volatility $v_{1,t}$ accounts for a small portion of the CDS spreads, and depreciation risk is the dominant component of the quanto CDS spreads.

Panels A–D in Figure 8 provide the time series of the decomposed USD-denominated CDS spreads and the depreciation and correlation risk components at one- and five-year maturities for the Eurozone countries. These panels also provide the time series of the two components of the quanto CDS spreads for the Eurozone countries at one- and five-year maturities for Period 1. Figure 8 depicts Period 1. During the period in which the CDS spreads are relatively higher, the common component with stochastic volatility constitutes a relatively higher part of the EUR-denominated CDS spreads than the idiosyncratic risk component based on the idiosyncratic risk process $z_t$ for Spain and Italy. By contrast, for Ireland and Portugal, the idiosyncratic risk components are the main drivers of local currency-denominated CDS spreads. Next, we turn to quanto spreads during the period in which the CDS spreads are relatively higher, and the correlation risk component constitutes a relatively greater part of the quanto spreads than in the rest of the sample period.

Additionally, at the five-year maturity, the proportion of the correlation risk component in the quanto CDS spread is greater than that for the one-year maturity.

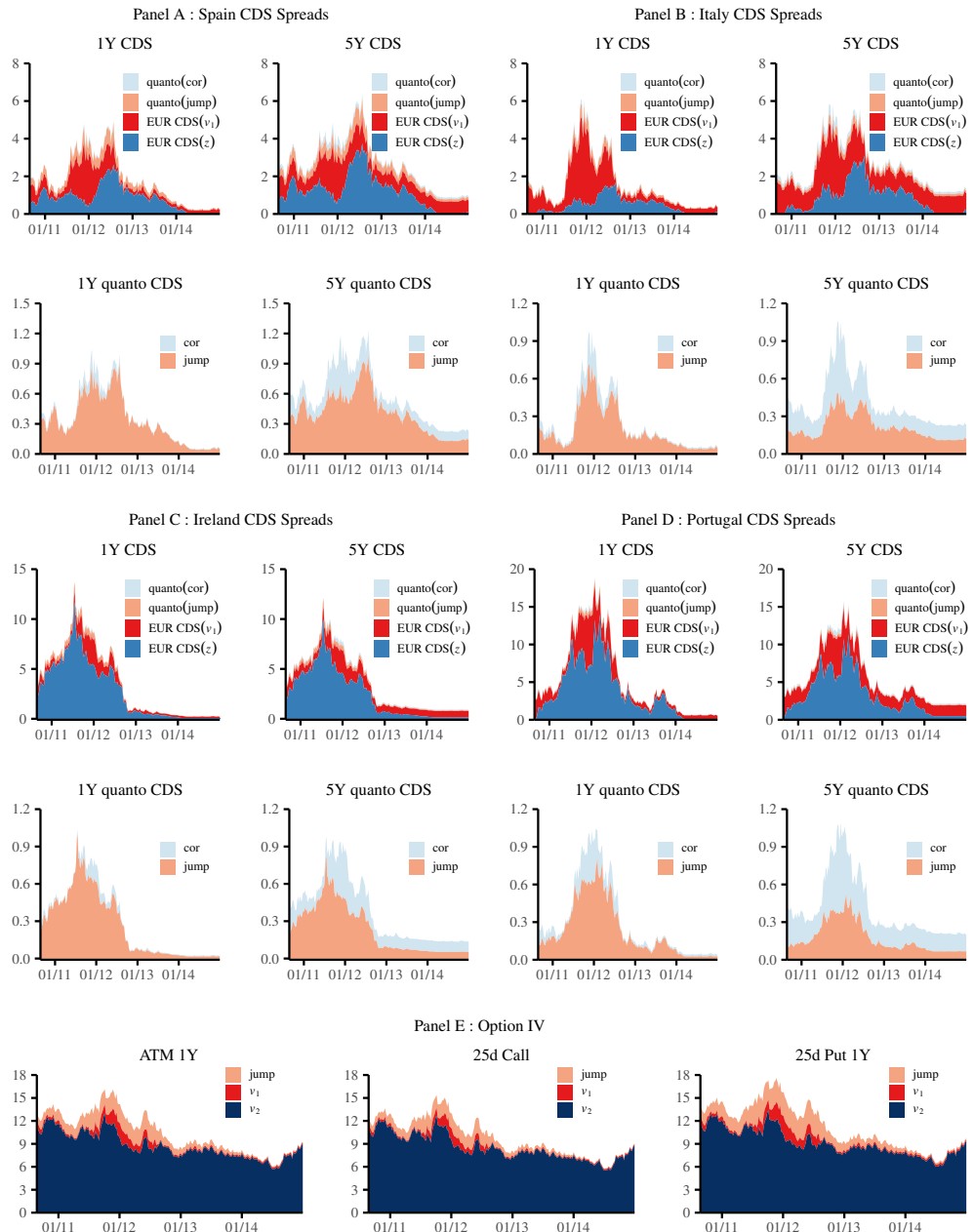

**Figure 8.** Decomposition of theoretical USD-denominated quanto CDS spreads and theoretical currency option implied volatility estimated by the two-factor stochastic volatility model for the Eurozone countries in Period 1. Panels A, B, C, and D show the decomposed components of theoretical USD-denominated CDS spreads and quanto CDS spreads for the Eurozone countries. The components consist of the idiosyncratic risk ($z_t$, blue) and common risk ($v_{1,t}$, red) of the EUR-denominated CDS spread, the depreciation risk (soft orange), and the correlation risk (light blue) at one- and five-year maturities for Spain, Italy, Ireland, and Portugal. Panel E shows the decomposed components of the theoretical currency option implied volatility for the EURUSD. The components consist of the idiosyncratic risk ($v_{2,t}$, dark blue), common risk, and depreciation risk at a one-year maturity. The sample period is from 25 August 2010 to 31 December 2014, with a weekly frequency. ATM: at-the-money.

Panels A–D in Figure 9 provide the time series of the decomposed components of the USD-denominated CDS spreads and quanto CDS spreads for Period 2 during the aftermath of the European sovereign crisis. Idiosyncratic risk and depreciation risks are the dominant components of the EUR-denominated CDS spreads and quanto CDS spreads, respectively.

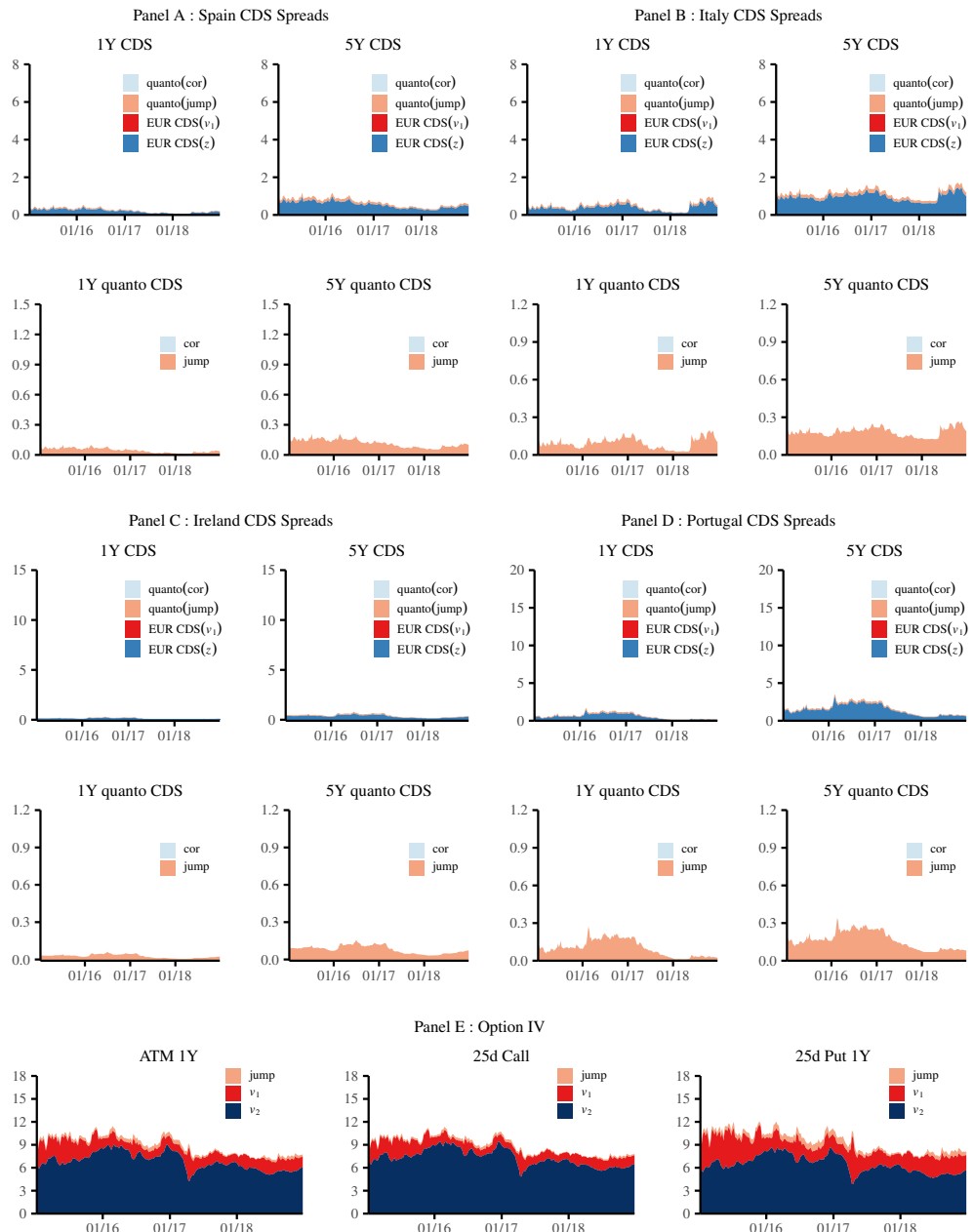

**Figure 9.** Decomposition of theoretical USD-denominated quanto CDS spreads and theoretical currency option implied volatility estimated by the two-factor stochastic volatility model for the Eurozone countries in Period 1. Panels A, B, C, and D show the decomposed components of theoretical USD-denominated CDS spreads and quanto CDS spreads for the Eurozone countries. The components consist of idiosyncratic risk ($z_t$, blue) and common risk ($v_{1,t}$, red) of EUR-denominated CDS spreads, depreciation risk (soft orange), and correlation risk (light blue) at one- and five-year maturities for Spain, Italy, Ireland, and Portugal. Panel E shows the decomposed components of the theoretical currency option implied volatility for EURUSD. The components consist of the idiosyncratic risk ($v_{2,t}$, dark blue), common risk, and depreciation risk at a one-year maturity. The sample period is from 7 January 2015 to 26 December 2018, with a weekly frequency. ATM: at-the money.

Panel E in Figure 9 provides the summary statistics of the implied volatility and its depreciation component of the currency option at one-month and one-year maturities for EURUSD. In Period 1, during the European sovereign crisis, the average and standard deviation of the depreciation risk component for the two-factor stochastic volatility model are lower than those for the one-factor stochastic volatility model. This finding is consistent with the estimated values of the expected depreciation ratios, which are lower than those of the one-factor stochastic volatility model during Period 1.

Furthermore, Panel E in Figure 8 provides the time series of the implied volatility process and the depreciation component of the currency option for EURUSD at one-year maturity in Period 1. Figure 10 provides the decomposed time series of the at-the-money implied volatility, risk reversal, and butterfly spread of the EURUSD currency option at the one-year maturity. For Period 1, the depreciation risk components increase the implied volatility levels by up to approximately 3% at one-year maturity and strengthen the negative slope of the volatility smile. The stochastic volatility $v_{1,t}$ component increases the implied volatility levels by up to approximately 2% at one-year maturity and strengthens the negative slope of the volatility smile. Because the correlation parameter $\rho_1$ is much higher than $\rho_2$, the stochastic volatility $v_{1,t}$ strengthens the negative slope of the volatility smile. In the currency option implied volatility, the effect of the depreciation risk is higher than that of the stochastic volatility $v_{1,t}$. For the euro expansion period, the depreciation risk effects are lower than those for Period 1. Panel E of Figure 8 shows the results of Period 2. The depreciation risk effects are lower in Period 2 than those in Period 1.

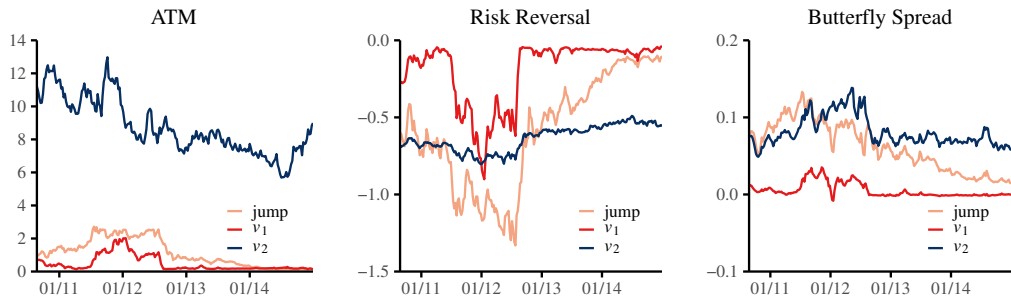

**Figure 10.** Decomposition of theoretical at-the-money (ATM) implied volatility, risk reversal, and butterfly spread estimated by the two-factor stochastic volatility model for EUR in Period 1. The currency option implied volatility of EUR is decomposed into the $v_1$ (red line), $v_2$ (dark blue), and depreciation risk (soft orange) components using the two-factor stochastic volatility model at one-year maturity. The sample period for Period 1 is from 25 August 2010 to 31 December 2014, with a weekly frequency. The units are percentages.

### 5.5. Risk Premiums

We next address the magnitude of risk premiums related to the risk factors. We follow Pan and Singleton (2008); Longstaff et al. (2011) and Augustin et al. (2020) and focus on the risk premium associated with unpredictable variation in stochastic volatility and idiosyncratic risk in the Eurozone during Period 1. First, we calculate the price based on the expectations under the physical probability measure.

$$S_t^{d,\mathbb{P}_d}(m) = \frac{(1-R)\int_t^{t+m} E^{\mathbb{P}_d}\left[e^{-\int_t^s r_u^d + \lambda_u^d du}\lambda_s^d \,\middle|\, \mathcal{F}_t\right]ds}{\Delta t \sum_{k=1}^{m/\Delta t} E^{\mathbb{P}_d}\left[e^{-\int_t^{t+k/\Delta t} r_u^d + \lambda_u^d du}\,\middle|\, \mathcal{F}_t\right]}. \tag{23}$$

$$c_t^{\mathbb{P}_d}(m) = E^{\mathbb{P}_d}[e^{-\int_t^{t+m} r_u^d du}(X_T - K)^+ | \mathcal{F}_t], \tag{24}$$

To quantify the risk premiums, we calculate the difference between CDS spreads (17) and (23) and currency options (19) and (24). Additionally, to confirm the impact of each risk factor, we calculate the difference assuming each $\eta_t$ equals 0. Table 8 provides the time series means of the risk premiums fractions $(S_t^d(m) - S_t^{d,\mathbb{P}_d}(m))/S_t^d(m)$ or $(IV_t(m) - IV_t^{\mathbb{P}_d}(m))/IV_t(m)$, where the option price $c_t^{\mathbb{P}_d}(m)$ is converted into the implied volatility of the option price $IV_t^{\mathbb{P}_d}(m)$. As shown, the average risk premium fractions are positive across all of the CDS spreads. This sign is consistent with the result of Pan and Singleton (2008) and Longstaff et al. (2011). This result is driven by the difference of the mean reversion parameter under $\mathbb{Q}_d$ and $\mathbb{P}_d$, as discussed in Section 5.1. The main driver of the risk premiums in the CDS spreads is the common risk $v_1$. A large part of the CDS spreads consist of these risk premiums. Additionally, the risk premiums driven by the common risk $v_1$ make the currency option implied volatility higher and the slope of the volatility smile steeper. However, the size of the risk premiums in the implied volatilities of the currency option are lower than those for the CDS spreads. The sign of the risk premiums driven by idiosyncratic stochastic volatility risk $v_2$ are negative and show an opposite pattern.

**Table 8.** Risk premiums estimated by the two-factor stochastic volatility model. The table shows the mean of the risk premium fraction of USD-denominated CDS spread at five-year maturity and currency option implied volatility at one-year maturity for the Eurozone countries of Spain (ES), Italy (IT), Ireland (IE), and Portugal (PT). The currency implied volatility is decomposed into a depreciation risk component. The sample period of Period 1 is from August 25, 2010 to December 31, 2014, with a weekly frequency.

| | | **USD CDS** | | | | **Currency Option** | | |
| | | **ES** | **IT** | **IE** | **PT** | **ATM** | **25-Delta Call** | **25-Delta Put** |
|---|---|---|---|---|---|---|---|---|
| $v_1$ | | 0.453 | 0.578 | 0.407 | 0.399 | 0.036 | 0.029 | 0.041 |
| $v_2$ | | 0.000 | 0.000 | 0.000 | 0.000 | −0.114 | −0.129 | −0.092 |
| $z$ | ES | 0.271 | 0.000 | 0.000 | 0.000 | 0.006 | 0.005 | 0.007 |
| | IT | 0.000 | 0.190 | 0.000 | 0.000 | 0.002 | 0.001 | 0.002 |
| | IE | 0.000 | 0.000 | 0.026 | 0.000 | 0.000 | 0.000 | 0.000 |
| | PT | 0.000 | 0.000 | 0.000 | 0.199 | 0.001 | 0.001 | 0.000 |

## 6. Conclusions

This study contributes to the literature on the relationship between different currency-denominated sovereign CDS spreads and currency options. We study the interaction between the term structures of sovereign CDS spreads and currency options through the Twin Ds.

We develop consistent pricing models for these instruments using a jump-diffusion stochastic volatility model, which allows us to decompose the term structure into risk components. These risk components include the exchange rate depreciation risk at a sovereign credit event, the correlation risk, and the common risk factor between the sovereign credit risk and the currency option market. We apply the models to different currency-denominated sovereign CDSs and currency options of the UK and Eurozone countries.

Unlike Carr and Wu (2007), following the result of the decomposition, the two-factor stochastic volatility model captures the dependence structure between sovereign CDS spreads and currency options for Eurozone countries during the European sovereign debt crisis. We find that, during the European sovereign debt crisis period, sovereign CDS spreads and currency options interact through the two risk components in the Eurozone countries. The intensity process of sovereign credit risk and the stochastic volatility of the exchange rate have a common risk factor. This common risk factor is the main driver of the CDS spread, especially in Italy and Spain and during periods of high market stress. During the same period, depreciation risk is the main driver of quanto CDS spreads and the negative slope of the volatility smile in currency options. The common risk factor also increases the currency option implied volatility and makes the negative slope of the volatility smile steeper. Specifically, the risk premiums driven by common risk factor

make these effects. The effect of the correlation risk in quanto CDS spreads is relatively low, unlike the findings of Lando and Bang Nielsen (2018), but it increases the quanto spread, especially at the five-year maturity. In contrast, the relationship between sovereign CDS spreads and currency options is weak in the UK, which has relatively low intensity levels.

Our results enhance our understanding of the price behavior between sovereign CDS markets and currency option markets for practitioners and policymakers. Our findings support the important interaction of the common risk factor and depreciation risk in both markets during periods of high market stress.

To understand the structure of the sovereign debt crisis, our findings have identified several key factors that influence the two markets. However, it is crucial to acknowledge the limitations inherent in our research and the avenues they open for future exploration. First, several studies (e.g., Aït-Sahalia et al. 2014 and Monfort et al. 2021) investigate the contagion risk in Eurozone sovereign CDS spreads. We do not consider the contagion mechanism in the pricing formulas and physical dynamics of sovereign CDS spreads. Incorporating the currency option or exchange rate into their models shows how credit risk affects the currency market as a contagion risk. Second, our model provides insight into the quanto effect for the instruments related to sovereign bonds. Sovereign bonds of the Eurozone countries are mainly denominated in EUR. The denominated currency is different from the USD-denominated CDSs, which are mainly traded for the Eurozone countries. Tsuruta (2020) investigates the relationship between the EUR-denominated sovereign bond yield and the USD-denominated CDS spreads, excluding our correlation effects. Incorporating EUR-denominated sovereign bond yield to our model enables us to evaluate a more accurate dependence structure in these instruments.

**Funding:** This research received no external funding.

**Data Availability Statement:** Restrictions apply to the availability of these data. Data were obtained from IHS Markit and are available with the permission of IHS Markit.

**Acknowledgments:** We wish to thank Nobuhiro Nakamura for his helpful comments on this article.

**Conflicts of Interest:** Author Masaru Tsuruta was employed by SBI Shinsei Bank, Limited. The author declares no conflict of interest.

## Appendix A. Solution for Sovereign CDS Spreads

First, we guess the expectation of premium leg in the foreign currency-denominated CDS pricing, as follows:

$$E^{\mathbb{Q}_f}\left[e^{-\int_t^{t+s}\lambda_u^f du}\middle|\mathcal{F}_t\right] = e^{-a^f(s)-b_1^f(s)v_t-b_2^f(s)z_t}.$$

$a^f(s)$, $b_1^f(s)$, and $b_2^f(s)$ satisfy the Riccati equations

$$\dot{a}^f = b_1^f\theta_v + b_2^f\theta_z, \quad \dot{b}_1^f = (1-\hat{\delta}_x^d)\beta - (k_v - \rho\sigma_v)b_1^f - \frac{1}{2}(b_1^f)^2\sigma_v^2, \quad \dot{b}_2^f = (1-\hat{\delta}_x^d) - k_z b_2^f - \frac{1}{2}(b_2^f)^2\sigma_z^2,$$

where $a^f(0) = 0$, $b_1^f(0) = 0$, and $b_2^f(0) = 0$. Next, we guess the expectation of the protection leg in the foreign currency-denominated CDS pricing as follows:

$$E^{\mathbb{Q}_f}\left[e^{-\int_t^{t+s}\lambda_u^f du}\lambda_{t+s}^f\middle|\mathcal{F}_t\right] = (1-\hat{\delta}_x^d)(c^f(s) + d_1^f(s)v_t + d_2^f(s)z_t)\phi,$$

where we denote that $\phi = e^{-a^f(s)-b_1^f(s)v_t-b_2^f(s)z_t}$. $c^f(s)$, $d_1^f(s)$, and $d_2^f(s)$ satisfy the Riccati equations:

$$\dot{c}^f = d_1^f\theta_v + d_2^f\theta_z, \quad \dot{d}_1^f = -(b_1^f\sigma_v^2 + k_v - \rho\sigma_v)d_1^f, \quad \dot{d}_2^f = -(b_2^f\sigma_z^2 + k_z)d_2^f.$$

where $c^f(0) = 0$, $d_1^f(0) = \beta$, and $d_2^f(0) = 1$. Therefore, we can solve these ordinary differential equations analytically, and the result is as follows:

$$
\begin{aligned}
a^f &= \frac{\theta_v}{\sigma_v^2}\left[2\log\left\{\frac{(\xi_v^f + (k_v - \rho\sigma_v))e^{\xi_v^f s} + (\xi_v^f - (k_v - \rho\sigma_v))}{2\xi_v^f} - (\xi_v^f + (k_v - \rho\sigma_v))s\right\}\right] \\
&\quad + \frac{\theta_z}{\sigma_z^2}\left[2\log\left\{\frac{(\xi_z^f + k_z)e^{\xi_z^f s} + (\xi_z^f - k_z)}{2\xi_z^f} - (\xi_z^f + k_z)s\right\}\right],
\end{aligned}
$$

$$
b_1^f = \frac{1}{\sigma_v^2}\frac{e^{\xi_v^f s} - 1}{\frac{e^{\xi_v^f s}}{\xi_v^f - (k_v - \rho\sigma_v)} + \frac{1}{\xi_v^f + (k_v - \rho\sigma_v)}}, \quad b_2^f = \frac{1}{\sigma_z^2}\frac{e^{\xi_z^f s} - 1}{\frac{e^{\xi_z^f s}}{\xi_z^f - k_z} + \frac{1}{\xi_z^f + k_z}},
$$

$$
\begin{aligned}
c^f &= \frac{2\beta\theta_v}{\xi_v^f + (k_v - \rho\sigma_v)} - \frac{4\beta\xi_v^f\theta_v}{\left\{(\xi_v^f + (k_v - \rho\sigma_v))e^{\xi_v^f s} + (\xi_v^f - (k_v - \rho\sigma_v))\right\}(\xi_v^f + (k_v - \rho\sigma_v))} \\
&\quad + \frac{2\theta_z}{\xi_z^f + k_z} - \frac{4\xi_z^f\theta_z}{\left\{(\xi_z^f + k_z)e^{\xi_z^f s} + (\xi_z^f - k_z)\right\}(\xi_z^f + k_z)},
\end{aligned}
$$

$$
d_1^f = \frac{4\beta(\xi_v^f)^2 e^{\xi_v^f s}}{\left\{(\xi_v^f + (k_v - \rho\sigma_v))e^{\xi_v^f s} + (\xi_v^f - (k_v - \rho\sigma_v))\right\}^2}, \quad d_2^f = \frac{4(\xi_z^f)^2 e^{\xi_z^f s}}{\left\{(\xi_z^f + k_z)e^{\xi_z^f s} + (\xi_z^f - k_z)\right\}^2},
$$

where $\xi_v^f = \sqrt{(k_v - \rho\sigma_v)^2 + 2\sigma_v^2(1 - \hat{\delta}_x^d)\beta}$, and $\xi_z^f = \sqrt{k_z^2 + 2\sigma_z^2(1 - \hat{\delta}_x^d)}$. Similarly, we guess the expectation of the premium leg in the domestic currency-denominated CDS pricing as follows:

$$
E^{\mathbb{Q}_d}\left[e^{-\int_t^{t+s}\lambda_u^d du}\bigg|\mathcal{F}_t\right] = e^{-a^d(s) - b_1^d(s)v_t - b_2^d(s)z_t}.
$$

$a^d(s)$, $b_1^d(s)$, and $b_2^d(s)$ satisfy the Riccati equations:

$$
\dot{a}^d = b_1^d\theta_v + b_2^d\theta_z, \quad \dot{b}_1^d = \beta - k_v b_1^d - \frac{1}{2}(b_1^d)^2\sigma_v^2, \quad \dot{b}_2^d = 1 - k_z b_2^d - \frac{1}{2}(b_2^d)^2\sigma_z^2.
$$

where $a^d(0) = 0$, $b_1^d(0) = 0$, and $b_2^d(0) = 0$. Next, we guess the expectation of the protection leg in the domestic currency-denominated CDS pricing as follows:

$$
E^{\mathbb{Q}_d}\left[e^{-\int_t^{t+s}\lambda_u^d du}\lambda_{t+s}^d\bigg|\mathcal{F}_t\right] = (c^d(s) + d_1^d(s)v_t + d_2^d(s)z_t)\phi,
$$

where $\phi = e^{-a^d(s) - b_1^d(s)v_t - b_2^d(s)z_t}$. $c^d(s)$, $d_1^d(s)$, and $d_2^d(s)$ satisfy the Riccati equations:

$$
\dot{c}^d = d_1^d\theta_v + d_2^d\theta_z, \quad \dot{d}_1^d = -(b_1^d\sigma_v^2 + k_v)d_1^d, \quad \dot{d}_2^d = -(b_2^d\sigma_z^2 + k_z)d_2^d,
$$

where $c^d(0) = 0$, $d_1^d(0) = \beta$, and $d_2^d(0) = 1$. Therefore, we can solve these ordinary differential equations analytically in the same way that the foreign currency-denominated CDS is solved.

## Appendix B. Solution for Currency Options

For currency option valuation for the one-factor stochastic volatility model, we can derive the generalized Fourier transform as follows:

$$
E^{\mathbb{Q}_d}\left[e^{iu\int_t^{t+s}(r_{d,u} - r_{f,u})du - a^{c1}(s) - b_1^{c1}(s)v_t - b_2^{c1}(s)z_t}\right].
$$

The ordinary differential equations below hold:

$$
\begin{aligned}
\dot{a}^{c_1} &= b_1^{c_1}\theta_v + b_2^{c_1}\theta_z, \\
\dot{b}_1^{c_1} &= \left\{(\zeta + iu\hat{\delta}_x^d)\beta + \frac{1}{2}(iu + u^2)\right\} - (k_v - iu\sigma_v\rho)b_1^{c_1} - \frac{1}{2}\sigma_v^2(b_1^{c_1})^2, \\
\dot{b}_2^{c_1} &= (\zeta + iu\hat{\delta}_x^d) - k_z b_2^{c_1} - \frac{1}{2}\sigma_z^2(b_2^{c_1})^2,
\end{aligned}
$$

where $\zeta = 1 - \exp(-iu\mu_j - v_j u^2/2)$, $a^{c_1}(0) = 0$, $b_1^{c_1}(0) = 0$, and $b_2^{c_1}(0) = 0$. Therefore, we can solve these ordinary differential equations analytically in the same way that the foreign currency-denominated CDS is solved. Given the above characteristic function, currency option prices can be calculated numerically via the COS method (Fang and Oosterlee 2009), which is based on the Fourier-cosine series for solving inverse Fourier integrals.

Next, for currency option valuation in the two-factor stochastic volatility model, we can derive the generalized Fourier transform as follows:

$$
E^{\mathbb{Q}_d}\left[e^{iu\int_t^{t+s}(r_{d,u} - r_{f,u})du - a^{c_2}(s) - b_1^{c_2}(s)v_{1,t} - b_2^{c_2}(s)v_{2,t} - b_3^{c_2}(s)z_t}\right].
$$

The ordinary differential equations below hold:

$$
\begin{aligned}
\dot{a}^{c_2} &= b_1^{c_2}\theta_{v_1} + b_2^{c_2}\theta_{v_2} + b_3^{c_2}\theta_z, \\
\dot{b}_1^{c_2} &= \left\{(\zeta + iu\hat{\delta}_x^d)\beta + \frac{1}{2}(iu + u^2)\right\} - (k_{v_1} - iu\sigma_{v_1}\rho_1)b_1^{c_2} - \frac{1}{2}\sigma_{v_1}^2(b_1^{c_2})^2, \\
\dot{b}_2^{c_2} &= \frac{1}{2}(iu + u^2) - (k_{v_2} - iu\sigma_{v_2}\rho_2)b_2^{c_2} - \frac{1}{2}\sigma_{v_2}^2(b_2^{c_2})^2, \\
\dot{b}_3^{c_2} &= (\zeta + iu\hat{\delta}_x^d) - k_z b_3^{c_2} - \frac{1}{2}\sigma_z^2(b_3^{c_2})^2,
\end{aligned}
$$

where $\zeta = 1 - \exp(-iu\mu_j - v_j u^2/2)$, $a^{c_2}(0) = 0$, $b_1^{c_2}(0) = 0$, $b_2^{c_2}(0) = 0$, and $b_3^{c_2}(0) = 0$. Therefore, we can solve these ordinary differential equations analytically in the same way that the foreign currency-denominated CDS is solved. Given the above characteristic function, currency option prices can be calculated numerically via the COS method.

## Notes

[1]  While there is no official period for the end of the European sovereign debt crisis, we illustrate the relationship between sovereign CDS spreads and currency option implied volatilities from 2009 to 2014 (see, e.g., Table 1), when the CDS spreads returned to their 2009 levels.

[2]  Several studies, including those by Bates (1996), Carr and Wu (2007), and Lando and Bang Nielsen (2018), use this assumption.

[3]  Several studies, including those by Bates (1996), Carr and Wu (2007), and Lando and Bang Nielsen (2018), use a jump-diffusion stochastic volatility model for the currency option volatility surface. Recently, Chernov et al. (2018) have considered the jumps in volatilities for a time-series analysis of daily exchange rate data. However, little attention has been given to the jumps in volatilities under a risk-neutral measure of the currency option volatility surface in empirical analysis. To give tractability to the changes in the two currency measures when deriving the formulas of different currency-denominated CDS spreads, we use a jump-diffusion stochastic volatility model.

[4]  As described in Section 3.1, we refer to the USD as the domestic currency and the currency of the country referenced by the sovereign CDS as the foreign currency. This is because the denominated currency of the benchmark sovereign CDS is the USD, and we use the dollar price of its currency for the currency option.

[5]  In previous CDS research, Longstaff et al. (2005); Pan and Singleton (2008) and Longstaff et al. (2011) use this model for CDS pricing.

[6]  Augustin et al. (2020) estimate the loss rate for sovereign CDS spreads of the Eurozone countries. Their estimated loss rate is 0.5925, so the implied recovery rate is $0.4075(= 1 - 0.5925)$.

[7]  We use CDS spreads with a complete restructuring clause. Mano (2013); Lando and Bang Nielsen (2018) and Della Corte et al. (2022) also use this clause.

[8]  Carr and Wu (2007) and Longstaff et al. (2011) use these same maturities. Liquidity differences across the maturities for sovereign CDS spreads are less than those for corporate CDS spreads (Augustin et al. 2020). Longstaff et al. (2011) argue that bid–ask

spreads of the one-year, three-year, and five-year contracts are reasonably similar, although the five-year contract typically has a higher trading volume.

[9] We include only four peripheral countries for two reasons. First, when we include many countries, the number of parameters increases, which also potentially increases the estimation time. Second, following the empirical result of the UK, whose intensity is lower than that of peripheral Eurozone countries, the correlation risk effect for quanto spreads and the depreciation risk for currency options are not relatively large.

[10] Bakshi et al. (2008) also use this conversion.

[11] According to Pan and Singleton (2008), this specification of the market price of risk is non-standard. The standard specification is $\psi_{i,0} = 0$. If $\psi_{i,0}$ is not zero, then the market price of risk is allowed to take on both a negative and a positive value. Pan and Singleton (2008) find that the flexibility of having non-zero $\psi_{i,0}$ and $\psi_{i,1}$ is essential for a reasonably good fit with the CDS data.

[12] Based on the simulation result, the authors show that the errors in this approach are negligible.

[13] This result is consistent with several studies' findings when estimating affine models; see, for example, Pan and Singleton (2008) and Zhang (2008) for emerging countries and Ang and Longstaff (2013) for the United States and European sovereigns. Ang and Longstaff (2013) also used the square-root process for the CDS spreads of European countries. Therefore, the intensity tends to be larger under a risk-neutral measure compared with a physical measure, and a significant negative risk premium is embedded in the pricing of the intensity. Market participants require a risk premium for exposure to variations in intensity.

[14] These authors also use the likelihood ratio test under the same assumption of state variables in the Kalman filter. They mention this same limitation.

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
