# Peer review of "Interaction between Sovereign Quanto Credit Default Swap Spreads and Currency Options"

_jrfm, doi:10.3390/jrfm17020085_

Round 1
Reviewer 1 Report
Comments and Suggestions for Authors
A well-written, relevant article, minor typos. Only real quibble is that many figures are far too small to be useful, but please see attached for more information. Good article.

Comments on the Quality of English LanguageQuality excellent - only some minor typos
Reviewer 2 Report
Comments and Suggestions for Authors
This paper studies the term structure of currency options and sovereign CDS quanto spreads, motivated by the expected risk of currency depreciation after sovereign credit default. The manuscript develops consistent pricing models by breaking down the term structure into risk components using a jump-diffusion stochastic volatility model. The study discovers a shared risk factor between the stochastic volatility of exchange rates and the intensity process of sovereign credit risk, which is the primary cause of the CDS spread, particularly in Italy and Spain, during times of intense market stress. It is also discovered that the primary cause of quanto CDS spreads and the negative slope of the volatility smile in currency options is depreciation risk. It is also argued that the common risk factor steepens the negative slope of the volatility smile and raises the implied volatility of currency options. In quanto CDS spreads, the correlation risk's impact is thought to be relatively low.
I believe that the chosen research topic is relevant and engaging enough to be taken into consideration for the Journal of Risk and Financial Management. The employed methodology appears to be sound from theoretical and econometric perspectives. The financial markets profession can benefit from fresh perspectives derived from this empirical research. Data was acquired from trustworthy sources over a sizable historical period (Bloomberg, IHS Market). Professional methods were used to conduct the empirical research.
My minor recommendations to raise the paper's caliber are as follows:
- In the Introduction section, position your paper in the context of academic literature. Write a summary of the key results and explain how the work adds to the body of literature. At the end of the introduction, also introduce the remainder of the paper's structure.
-A literature review ought to be designed with the research objectives and paper findings in mind, rather than just summarizing the findings of the articles it has reviewed.
-Update your reference list with sources that are comparatively more recent (from 2019 onward).
-Write your study's implications in the Conclusion section. Determine the limitations of your empirical research, if any, and establish the goals for further investigation to carry out this study.
